

# Earthquake static stress transfer in the 2013 Valencia Gulf (Spain) seismic sequence

Lluis Salo[1,2], Tanit Frontera[2], Xavier Goula[2], Lluis Pujades[1], Alberto Ledesma[1]

[1]Department of Civil and Environmental Engineering, Universitat Politecnica de Catalunya, Barcelona, 08034, Spain
[2]Institut Cartografic i Geologic de Catalunya, Barcelona, 08038, Spain

*Correspondence to*: Lluis Salo (lluis.salo@upc.edu)

**Abstract.** On September 24[th], 2013, a $M_L$ 3.6 earthquake struck in Valencia Gulf (Spain), near the Mediterranean coast of Castellon, roughly a week after the gas injections conducted in the area to develop an Underground Gas Storage had been halted. The event, felt by the nearby population, led to a sequence build-up of felt events which reached a maximum of $M_L$
4.3 on October 2[nd].

Here, we study the role of static stress change as an earthquake triggering mechanism during the sequence, and provide quantitative assessment of the known faults final stress state. By means of the Coulomb Failure Function, the evolution of static stress is quantified both on fault planes derived from focal mechanism solutions (which act as source and receiver faults), and on the previously mapped structures in the area (receiver faults). Results show that static stress transfer could
have acted as a partial trigger, and point towards an ESE-dipping structure as the most likely to have been activated during the sequence. Based on this approach, the influence of the studied events in the occurrence of future and potentially damaging earthquakes in the area would be, at most, of second order.

## 1 Introduction

With an increasing demand of energy resources, the Spanish government ideated the CASTOR Underground Gas Storage (UGS) project in 1996 (Fig. 1a). Its development should have provided more autonomy to the Spanish gas system, highly
dependent on incoming gas from northern Africa and Europe. The selected geological structure to store gas (Fig. 1b) was the depleted Amposta Oil field which had been operated by Shell in the rough period 1970-1990, exploiting its naturally contained heavy oil of 17 ° API (Seeman et al. 1990; Batchelor et al. 2007; Escal 2014a; Escal 2014b).

Natural seismicity in the area is very low (Instituto Geologico y Minero de España, IGME, 2015a,b), despite the fact that several faulting structures had been described or inferred by previous studies in the surroundings of the UGS, mainly using a



geological approach (Fontbote et al., 1990; Roca and Guimera, 1992; Gallart et al., 1995; Verges and Sabat, 1999; Perea, 2006; IGME, 2015a,b). ESCAL, the operating society of the UGS, obtained a 3D cube of seismic reflection profiles which provided detailed information in the reservoir surroundings (Geostock, 2010). The reservoir itself is located roughly in between 1.7 and 2.5 km beneath the seabed. The fault system derived from Geostock's (2010), as appearing in Cesca et al.

(2014), is considered here (Fig.1c).

During the third injection phase of cushion gas, carried out from September 5th to 17th, 2013, seismic activity built up to more than 30 events per day (Fig. 2). Although the immediate response after injections halt on September 17th was a decrease in seismic activity, on September 24th the first felt event took place ($M_L$ 3.6), and on October 1st and 2nd, while near one-hundred events were being recorded per day, three events of $M_L > 4$ occurred, raising public concern. The last day with felt

events was October 4th, and seismicity returned to basal levels at the end of the same month. After the experienced earthquakes, the project was paused and is at present on hibernation phase. The Spanish government has to decide whether the facility should be dismantled or storage operations can be resumed (e.g. Gonzalez, 2014).

Non-confidential literature in relation to the CASTOR case has already addressed problems in earthquake locations and focal mechanisms' computation, as well as the frequency-magnitude relationship (Institut Cartografic i Geologic de Catalunya,

ICGC, 2013; IGME, 2013; Instituto Geografico Nacional, IGN, 2013; Cesca et al., 2014; Gonzalez, 2014; Gaite et al., 2016). Cesca et al. (2014), included some discussion in relation to the mapped faults which would have been more likely to slip based on background stress as well. However, quantification of the physical mechanisms that may generate earthquakes and its evolution in the sequence had not been done beforehand. In this study we aim at providing new evidence on the influence of a specific earthquake-triggering mechanism, known as static stress transfer (King et al. 1994), during the seismic

sequence. Static earthquake triggering can be understood as the result of a fault slip (strain) which, accounting for a confined medium, translates into a stress perturbation which can destabilize other faults.

Owing to earthquake location results and uncertainty (e.g. Gaite et al., 2016), and given the geologic structure complexity in the area, directly relating earthquakes with known faults should not be attempted. The approach followed here uses information derived from Focal Mechanism (FM) solutions of the 8 main events in the sequence ($M_L$ 3.5 and above) in order

to place and model geometric characteristics of source faults. Coulomb Stress Changes ($\Delta$CS) are studied both on these planes as the sequence evolves, and on the previously mapped structures (Fig. 1c), the latter being treated as receiver faults only. Therefore, we work with the hypothesis that the studied earthquakes could be the result of slip along fault planes which had not been described previously. This is in accordance with both FM solutions obtained here (see sect. 2.3), and Gaite's et al. (2016), paper findings.



## 2 Method

### 2.1 Coulomb failure function

Although various criteria have been used to determine failure conditions on rocks (e.g. Jaeger and Cook, 1979), the Mohr-Coulomb failure criterion is probably the more widely used. It is defined as shown in Eq.( **1** ):

$$\tau = c' + \mu\sigma' \tag{1}$$

Where $\tau$ is the failure shear stress that is related to a particular value of the effective normal stress $\sigma'$ acting on the plane, given a coefficient of friction $\mu$ and an effective cohesion $c'$. The Coulomb Failure Function (CFF), which allows obtaining the $\Delta$CS, can be understood as the sum of the shear stress change (always positive and thus, destabilizing) and the frictional term (positive when the fault is unclamped; e.g. Lin and Stein, 2004; Toda et al., 2005), which results from moving the frictional term in Eq.( **1** ) to the left of the equal sign, and it is expressed as in Eq.( **2** ). As the focus is on the stress change,

the cohesive term of the Mohr-Coulomb failure criterion is omitted and the equation is formulated using incremental terms $\Delta$.

$$\Delta CS = \Delta\tau \pm \mu\Delta\sigma' \tag{2}$$

In order to account for the effect of pore pressure increase, Eq.( **2** ) can be rewritten as in ( **3** ) using ( **4** )(see Sumy et al., 2014, for more detail):

$$\Delta CS = \Delta\tau \pm \mu'\Delta\sigma \tag{3}$$

$$\mu' = \mu(1 - \beta_k) \tag{4}$$

Where $\mu'$ and $\beta_\kappa$ are the effective friction and the Skempton coefficients, respectively. Hence, it is the effective coefficient of friction ($\mu'$) that incorporates the fluid pressure effect. Besides the fact that hydrogeologic parameters of the faults (affected

by fluid injections) are not published and therefore unknown to the authors, knowing the exact value of $\mu'$ on a particular fault is a matter of ongoing research and was not attempted here. But, it has long been acknowledged to range from 0.0 to 0.8 (e.g. Parsons et al., 1999; Sumy et al., 2014). Its recommended value in calculations for strike-slip or unknown faults is 0.4 (friction $\mu = 0.75$, Skempton's coefficient $\beta_\kappa = 0.47$), as it minimizes the maximum associated error if the previously defined limits are taken into account (Stein et al., 1992; Toda et al. 2011a).

To deal with tridimensional complexity in calculations we used the COULOMB 3.3 software (Toda et al., 2011b), which presupposes an homogeneous elastic halfspace and implements Okada's (1992) solutions to compute strains. Bearing in mind that all the analysis is performed at depths of 11 km or shallower, the assumption of an homogeneous crust density model was justified (Diaz and Gallart, 2009).

### 2.2 Shortening of the seismic cycle

The characteristic earthquake theory accepts that faults will slip with a series of identical (characteristic) earthquakes (Scholz, 2002). The process of strain accumulation and release on a fault is a cyclical process that takes place with a



recurrence time $T_r$, and so the seismic cycle can be defined. The introduction of an external perturbation, such as the one caused by fluid injection or previous earthquakes, may accelerate the process (Fig.3).

Therefore, the shortening of the seismic cycle $\delta_{cyc}$ can be quantified as in Eq.( 5 ), for an associated earthquake stress drop $\Delta\sigma$ and $T_r$ (Harris, 2000; Baisch et al., 2009). However, results can be given in terms of the relative reduction (e.g. %), without

introducing the $T_r$ and increasing uncertainty. Here results using both will be presented.

$$\delta_{cyc} = \frac{\Delta CS}{\Delta\sigma} T_r \tag{5}$$

The earthquake stress drop can be calculated as in Eq.( 6 ), by relating it to the average strain change ($\frac{\Delta u}{\hat{L}}$) using Hooke's law (Lay and Wallace, 1995).

$$\Delta\sigma = C \cdot G \left(\frac{\Delta u}{\hat{L}}\right) = C \left(\frac{M_o}{Rupt.\,Area \cdot \hat{L}}\right) \tag{6}$$

$$C\ (strike\ slip) = \left(\frac{2}{\pi}\right)\ ;\ C\ (dip\ slip) = \left[\frac{4(\lambda + G)}{\pi(\lambda + 2G)}\right]$$

Where in the context of elasticity, $G$ is the shear modulus or Lame's second parameter,  and $\lambda$ is Lame's first parameter. Under usual crust conditions, both are roughly of the order of 3E+10 Pa. Here we compute both strike-slip and dip-slip stress

drops based on the assumed fault dimensions, but maximum and minimum probable values (3 and 1 MPa respectively) are introduced as well (Beeler et al., 2000; Baisch et al., 2009). Following Eq.( 5 ), if larger stress drops are considered (e.g. Kanamori and Anderson, 1975), the influence of the computed $\Delta CS$ value is smaller (the ratio $\Delta CS/\Delta\sigma$ diminishes); hence, it would be a non-conservative hypothesis, as the obtained $\delta_{cyc}$ values would be minor.

On the other hand, the $T_r$ can be estimated as the ratio between the expected seismic moment $M_o^e$, which can be obtained from

the moment magnitude $M_w$ using Hanks and Kanamori's (1979) relation, and the geologically assessed moment rate $M_o^g$ (Wesnousky, 1986; Perea, 2006). The formulae needed can be seen in Eqs. ( 7 ) to ( 9 ). In Eq.( 8 ), $A$ is the area of the fault and $SR$ is the slip rate. SR values are only reported in literature for the Main Fault (Perea, 2006; Garcia-Mayordomo et al., 2015).

$$M_o^e = 10^{1.5(M_w+10.7)}\ [dyn \cdot cm] \tag{7}$$

$$M_o^g = G \cdot A \cdot SR \tag{8}$$

$$T_r = \frac{M_o^e}{M_o^g} \tag{9}$$

## 2.3 Fault modelling

### 2.3.1 Focal mechanism computation



We used Delouis' (2014) method FMNEAR to determine FM solutions, via its online webservice (Delouis et al., 2016). The process to obtain the FM solution consists on inverting the waveform and obtaining the double couple of forces, whose information is contained in the Seismic Moment Tensor (SMT). Such a thing can be done thanks to the linear relationship between ground motion and the components of the SMT, which allows obtaining the latter via linear inversion. From the
SMT, fault geometry is deduced and so a FM solution is obtained.

Apart from the waveforms recorded at each station, the program needs as an input the earthquake location and an initial magnitude estimate (see sect. 3). The output file given by FMNEAR provides the intrinsic quality of the waveform modelling (RMS misfit and visual comparison) and a Confidence Index (CI), which is an indicator of the reliability of the solution, in between 0 and 100. Based on Delouis' (2014), a FM solution can be considered trustworthy when its CI is above 70.

The selected velocity model to perform calculations is shown in Table A1 (Appendix A). It corresponds to the default model in FMNEAR webservice, which accounts for an average crust and has been proven to  provide reliable results in various geological contexts (Delouis, 2014).The used FM are those corresponding to the 8 main events of the sequence, ranging from $M_L$ 3.5 to 4.3 (see Fig.4), as no reliable solutions could be obtained for smaller events. They are presented in table format as well, showing all relevant parameters (Table B1 in the Appendix B). Solutions correspond to strike-slip mechanisms with
some normal component, which is in clear agreement with previous findings by Frontera et al. (2013), and IGN (2013), and similar to Cesca et al. (2014), results.

### 2.3.2 Source faults

Each FM solution provides two nodal planes, of which only one corresponds to the source mechanism. Although it is not possible to distinguish between both *a priori*, the usual procedure consists in comparing each option with known faults in the
area and then deciding which plane is feasible and which one is not. As seen in Table B.1, best solutions' depth ranges from 5 to 8 km for most of the events. This is in accordance with Gaite et al. (2016) results, which would place most of the events at a depth of about 6 km, but cannot be related with current detailed geological knowledge of the area, as it does not reach depths beyond 3 km (IGME, 2013; Cesca et al., 2014; Fernandez et al., 2014; Garcia-Mayordomo et al., 2015).

Hence, and considering static stress transfer as a potential driving mechanism during the sequence, the selected nodal plane
of each FM solution was inferred from a test-and-error analysis during the sequence of the 8 comprised events, assuming that the causative fault plane was the one with higher computed ΔCS. This means that after each earthquake, the ΔCS was checked on both nodal planes of the FM solution of the following one, and the one with greater values was chosen as the hosting plane of that next event.

Fault plane size is modelled using Wells and Coppersmith (1994), relations, and faults are centred at the location of each
earthquake. Due to small area *A* of the planes (roughly 1 x 0.5 km for a $M_w$ 4 rupture), we model source faults using only one patch. As ΔCS is resolved on each of the 8 causative planes after each event, these planes act as source faults when they slip, but also as receiver faults along the sequence. Slip values corresponding to each event's dislocation Δ*u* are tested using Eq. (



**10** ) to obtain the resulting seismic moment (e.g. Lay and Wallace, 1995), and are varied until the $M_o$ corresponds to the $M_w$ of each event using the well-known Hanks and Kanamori (1979), formula.

$$M_o = G \cdot A \cdot \Delta u \tag{10}$$

### 2.3.3 Receiver faults

Taking into account the reasons given above, mapped faults near the CASTOR UGS only receive stress variations due to the

source faults' slip. In order to generate the input file for COULOMB calculations, a geometrical 3D model of these receiver faults was first created, using information in  Seeman et al. (1990), Batchelor et al. (2007), Playa et al. (2010), IGME (2013), Cesca et al. (2014), ESCAL (2014a,b), Fernandez et al. (2014), and Garcia-Mayordomo et al. (2015). Fig. 5a shows two horizontal slices of the model.

The listric morphology is simulated in COULOMB using a three-patch assembly along the vertical dimension of the fault,

each one with a different dip angle and according to descriptions in the literature mentioned in the previous paragraph. Divisions along the base of the fault depend on its length (2 to 8 patches), resulting in receiver faults being divided in a total number of patches that ranges in between 6 and 24. Fault traces, which if looked in detail appear curved, were digitized to a straight line maintaining the average strike direction. Rake of the receiver faults has been inferred from FM information (as it should accord with background stress regime), the decided value being the average of the FM solutions' rakes. Of the two

nodal planes provided by each solution, the selected nodal plane to average is that closer to each fault's strike. However, regarding the Main Fault both were tested and the most adverse (i.e. higher values of resulting ΔCS) was used, to provide a conservative approach from a seismic hazard standpoint. The described model for calculations is shown in Fig. 5b.

### 2.4 Dealing with uncertainty

The assumption that the theory of elasticity is acceptable helps in reducing the amount of parameters needed and their

associated errors. However, uncertainty in some of them (e.g. the effective friction coefficient μ') should be taken into account. To manage this matter appropriately, a sensitivity test of the final result computed on the receiver faults has been carried out. Variations have been introduced in the parameters affecting the CFF (strike, dip, rake of source and receiver faults and μ') but always maintaining the nature of FM solutions (for example, rake is not varied more than ± 20º from the FM solution value). This helps in handling the simplified fault model (fault traces modelled as straight lines) as well.

The robustness of the result is finally addressed by varying the depth of source faults. Changes in the final stress state are explored on the Main Fault, as it is the structure mainly contributing to seismic hazard in the area. The focus is on the spatial extension (total area) of loaded patches. Both the $T_r$ and $\Delta\sigma$ are considered within a range and not a unique specific value as well. Conversely, both a reliable value of the Young's modulus and Poisson coefficient have been established for studies of

this kind (e.g. Toda et al., 2011b). Here we use 8E+10 Pa and 0.25 respectively. A summary of all parameters and their chosen ranges in the performed analysis are shown in the Appendix C, table C1.

## 3 Data and resources

To compute FM solutions, which are then needed to generate the source faults, a reliable catalogue of earthquake locations,
magnitudes and the corresponding recorded waveforms were used. Here we chose the published seismic bulletin in ICGC (2015), as well as the recorded waveforms by the seismic stations shown in Fig. 1a. As an example, Fig. 6 shows the FM and the waveform fit at stations ERTA and CMAS for the M 3.9 event of 09/30, as obtained by applying Delouis' (2014) method via FMNEAR webservice (Delouis et al., 2016; see sect. 2.3.1). On the other hand, receiver faults are modelled according to the existing references (see sect. 2.3.3).
Throughout the whole analysis, we keep an eye on hypocentral locations reported by Cesca et al. (2014), and Gaite et al. (2016), in addition to those in ICGC (2015); when it comes to the interpretation of the results as a whole, the distribution of the earthquake cloud (including those events with M < 3.5) should not be left aside.

## 4. Analysis result

### 4.1 Static stress variation on the source fault planes

Firstly, the analysis checks the evolution of ΔCS in the causative planes of the occurred sequence, after each event introduces stress changes in the area which are combined to those of the previous events. The sequence is represented here by means of the 8 main events for which a reliable FM solution was obtained, as shown in Fig. 7. All earthquakes but one (09/30), had positive ΔCS on their nodal planes before they occurred. The quantitative variation is better appreciated in the time series depicted in Fig. 8. Except for the event of 09/30 and the first of 10/04, this latter having a slight decrease of Coulomb stress
on it before slipping, all other planes slipped not only when ΔCS were positive, but also when they were at their highest. As expected due to earthquake magnitudes and distance, positive values are small (maxima around 0.1 bar), but variations of this order of magnitude have already been related with locations where aftershocks occur (Stein, 1999; Mulargia and Bizarri, 2014).

### 4.2 Static stress variation on the mapped fault planes

The study explained in the previous section is performed on the receiver fault planes as well (the previously mapped faults near the CASTOR UGS). Fig. 9 and Fig. 10 are time series on the mapped fault planes showing the variation on all modelled patches along the sequence. The final stress state can also be seen on a model excerpt in Fig. 11. Whereas the East 2 fault and





the Montsia system faults have near 0.0 positive values, larger changes are obtained on both the Main (certain patches) and East 4 faults.

On one hand, near 0.15 bar are achieved on a southern patch of the Main Fault, which has a remarkable number of patches left with positive values at the end of the sequence. Regarding the East 4 fault, two important stress drops are observed after

the main shock ($M_w$ 4.2 on October 2[nd], Julian Day 275) and after the last event of October 4[th]. Following the 8 considered events, this fault is globally unloaded in terms of ΔCS, as depicted in Fig. 11 (Minimum value resolved onto one of its patches being -0.25 bar). Hence, should one of the mapped faults have been activated during the sequence, East 4 is the most likely to have slipped.

### 4.3 Sensitivity

**4.3.1 μ' and fault geometry**

Fig. 12 shows how the final result on each receiver fault changes when variations in the geometrical parameters of the source faults and effective friction are introduced. The maximum and minimum values resolved on a patch, as well as the average value (mean of the ΔCS at each modelled patch, for every fault) are shown. Logically, the greater the final ΔCS, the higher the absolute change when any variation is introduced as well, and that is why both the East 4 (minimum values) and Main

Fault (maximum values) register more variation in the final result when a parameter value is changed. As noted in sect. 2.4, all parameter variations are summarized in Table C1. Based on the obtained result, for a ± 10 ° variation in the geometrical parameters, the model is most sensitive to fault orientation (strike), which should be expected (King et al., 1994; Note, in Table C.1, that rake is moved up to ± 20 °).

Average values help in contextualizing whether the fault, as a whole, would have been left with an unstable stress state. It can

be seen that, unlike maximum and minimum values, the average value is practically left unchanged in all faults other than the East 4, regardless of the variation introduced. Furthermore, average values never exceed ± 0.1 bar except for the East 4 fault once again. First, average values remaining near constant shows the robustness of the final result. And second, the fact that the mean value in the East 4 fault is, for most of the introduced variations, near (or even exceeds) -0.1 bar, supports the plausible idea of this structure to have slipped, initially derived from the stress drops observed in sect. 4.2.

A strike direction change in the receiver faults themselves (e.g. due to the curved faults' trace) is already covered by the variation of strikes in the source faults (the relative variation of leaving the source fault's strike unchanged and moving the strike in any receiver fault is analogous). However, dips and rakes of the receiver faults were also varied within the ranges shown in Table C1, obtaining no greater differences than the previously presented. No sensitivity study was undertaken regarding the final result in the source faults (derived from FM solutions), as they represent unknown structures. Moreover,

given the dimensions used and their stress states after the considered earthquakes, the corresponding fault areas should have been left far from slipping again.



### 4.3.2 Focal depth

Due to seismic gap, minimum distance hypocenter-station (in the range 20-30 km; Fig. 1a) and an 1D velocity model used to perform locations (ICGC, 2015), depth is the spatial dimension subject to greater uncertainty (in the range of ± 3 km for most of the solutions). Consequently, even the locations for the strongest events in the sequence suffer from depth uncertainty

which should be accounted for (e.g. Cesca et al., 2014; Gaite et al., 2016). With this idea in mind, the final result when it comes to ΔCS onto the Main Fault was checked by moving source faults' depth 1, 2 and 3 km upwards, until the shallowest focus reached surface depth (refer to tables B1 and C1). Switching source depths to shallower points affected in positive perturbations being of greater magnitude (distances are minor), but most of the fault area was either unloaded or unaffected in terms of ΔCS. Therefore, a location depth error up to 3 km on the studied earthquakes does not remarkably change the

total area of the Main Fault subject to positive ΔCS, and thus its influence when it comes to the shortening of the seismic cycle of future characteristic earthquakes should be minor.

### 4.4 Relating the studied sequence to the occurrence of future earthquakes

Tables 1 and 2 show the results computed as shown in Eq. 5, regarding the Main Fault. The considered magnitudes of the earthquakes for which the influence of the studied sequence is to be quantified are 6.0, 5.0 and 4.5. The first corresponds to

the characteristic magnitude (Schwartz et al., 1981; Schwartz and Coppersmith, 1984) taking into account the total modelled area. The last two correspond to moderate shakes which could be expected in the area with relatively common occurrence (e.g. hundreds of years), based on known dimensions of present structures.

Results are plotted for the "best estimate" of input parameters (μ' = 0.4 and source parameters according to FM solutions), and assuming a range up to 4 different values of the stress drop, according to the reasons given in sect. 2.2. Moreover, the

worst tested assumption for each parameter change (strike, dip, rake, μ' and focal depth) is used as well. Due to the number of stressed patches, the mean value of the ΔCS is used when taking into consideration the $M_w$ 6.0 earthquake (for which all fault area is expected to slip; assumed rupture dimensions following Wells and Coppersmith, 1994), while the maximum positive value is selected for both the $M_w$ 5.0 and 4.5 events.

Regarding the characteristic earthquake, $\delta_{cyc}$ never exceeds 0.5 %. It corresponds to a maximum advancement in the

occurrence of 22 y. out of the 5191 y. estimated, although $T_r$ values should be treated very carefully as uncertainty is noteworthy. Shortening values around 5 % could be attained for moderate events of lower magnitudes. Besides, it was observed that even introducing the maximum positive value resolved on an individual patch into calculations, $\delta_{cyc}$ would not exceed 6 % for the $M_w$ 6.0 earthquake. Results owe their values to small ΔCS when compared with the associated stress drops.

### 5 Discussion



## 5.1 Static stress transfer as a triggering mechanism

Out of each pair resulting from a FM solution, we used the assumption that the slipping planes were the ones with greater ΔCS resolved onto them, and so, when addressing the question of the weight of the considered trigger in the sequence, it is logical to ask how our results would have been affected, had the other plane been taken as the causative plane. As explained

in sect. 2.3.2, the plane was selected by test-and-error, which means that both were already checked for each solution, and it was found that picking the other nodal plane did not change the resolved ΔCS nature (being positive or negative), but just their magnitude, something usually found in calculations (e.g. Sumy et al., 2014). As noted before, earlier studies have shown that ΔCS values as little as 0.1 bar can promote seismicity; but, as well, most of them would advocate for areas with resolved ΔCS being smaller than 0.1 bar neither having seismic activity promoted nor inhibited (e.g. Reasenberg and

Simpson, 1992; Harris, 2000). Hence, it is interesting to note that although this study shows positive values of ΔCS found in 6 out of 7 source planes, only the nodal planes of the last 3 events had computed values near the mentioned threshold (Fig. 8).

In order to provide further insight, Fig. 13a depicts the orientations of Optimally Oriented Fault Planes (OOFP) superposed to the source and receiver faults' traces. We assume a current strike-slip stress regime with main horizontal stress ($S_H$)

orientation about 15 ° N, derived from our FM solutions and previous studies (Schindler et al., 1998; Heidbach et al., 2008; Cesca et al., 2014). Stereographic projections of all nodal plane solutions from FM on a polar sphere are shown as well (Fig 13); note that they essentially correspond to 2 different possibilities. The orientations of the two nodal plane families (NW-SE and NNE-SSW), are comparable to those of the OOFP (strike-slips), which are found at roughly ± 30 ° with respect to the $S_H$ (King et al., 1994; Zoback, 2007). This means that both FM nodal planes are very well-oriented in relation to the regional

stress field. On a critically stressed crust (e.g. Stein and Lisowski, 1983; Bak and Tang, 1989), small-magnitude destabilizing stress changes may be enough to initiate a rupture that can propagate along a fault's total area, even to those regions where the external perturbation does not surpass the fault's friction-based shear resistance, and generate felt events if dimensions allow it (Gischig, 2015; Piris et al., submitted). In addition, the reader should note that pore pressure generation cannot be accounted for using Coulomb static stress modelling, as well as other earthquake-triggering mechanisms that might be

present. Therefore, it seems coherent that static stress acted as a partial trigger along the sequence, interacting with other physical processes that can destabilize faults.

## 5.2 The hosting faults

Time series study on currently mapped faults showed two remarkable stress drops resolved onto the East 4 fault plane, thus unloading the fault and preventing a new rupture after the considered events. According to Wells and Coppersmith (1994), a

magnitude ~ 4 earthquake is assumed to be the result of a rupture along a 0.5 km² area, which is less than the modelled area for each individual patch in the East 4 fault. Thus, the movement of only one patch (possibly any of the unloaded ones) could



potentially produce an event in the magnitude range of the studied events already. Moreover, both the sensitivity analysis shown in Fig.12 (remarkable negative peak and average values) and the OOFP revision discussed in the previous section provide partial evidence that a scenario in which the East 4 slipped one or more times to produce (part of) the studied earthquakes is plausible.

Cesca et al. (2014), whose locations were resolved at 1-3 km beneath the seabed (around reservoir depth), reported that based on the current stress field an ESE dipping fault was better oriented and so its probabilities of being the source fault were higher. This result is in accordance with ours, other than the depth difference of the resolved FM solutions. As noted from the beginning, depth uncertainty cannot be very well constrained here because of the minimum distance, existing gap and crust velocity model. However, Gaite et al. (2016), used a new 3D shear-wave velocity model derived from surface wave ambient

noise tomography, which is probably more accurate than the CRUST 2.0 average model (Bassin et al., 2000) used by Cesca et al. (2014). From their findings it is clear that focal depths of the CASTOR sequence could mostly be beneath reservoir (and known faults) depth, at ~ 6 km. Hence, while the East 4 is visibly the most likely to have been activated among the mapped faults, the hypothesis of greater focal depths and so unknown faults being present in the area cannot be discarded. Conversely, no evidence supporting a rupture along the Montsia system nor the Main Fault was found, thus confirming

previous results (IGN, 2013; IGME, 2013; Cesca et al., 2014). Both were first discarded from the FM solutions, which show a NW-SE striking plane which is almost vertical, while the Montsia system has a gentler dip (about 50 °), and a NE-SW oriented plane which dips towards the East (the Main Fault dips to the NW). Additionally, the performed $\Delta$CS study points towards the same idea.

**5.3 Could the occurrence of future earthquakes have been shortened as a result of static stress transfer?**

The reduction of the seismic cycle $\delta_{cyc}$, as a result of the studied sequence and regarding 3 different earthquakes on the Main Fault ($M_w$ 4.5 - 6.0), was found to be minor (5 % for the moderate shakes and less than 1 % for the characteristic earthquake). Although an estimation of the return period $T_r$ of an earthquake was not possible for the other shown faults (unknown slip rates), magnitudes of the resolved $\Delta$CS make it unlikely for the corresponding seismic cycles to have been shortened to a greater extent, provided the associated stress drops are similar (see Eq.( **5** )). Considered stress drops are in the range 10-40

bar, and so the computed $\Delta$CS would have to increase at least an order of magnitude to have a relevant influence (i.e. shortening of 10 % or greater).

Such a raise could have been found if source faults had been shallower, but destabilizing perturbations were found to always be very localized for those ruptures. This means that while a significant shortening could potentially have been found for moderate events (up to $M_w$ 5.0), had the earthquakes struck at shallower depths, the probability that most of the Main Fault's

area had been stressed to relevant levels (promoting failure) is very low. Thus, all evidences in this study indicate that the weight of the studied sequence in the occurrence of a strong earthquake ($M_w$ 6.0) can be neglected.





## 5.4 Shortcomings

While the present study allows quantifying a specific earthquake trigger to assess its influence in the sequence and future occurrence of earthquakes, it cannot rigorously address the issue about the origin of seismicity. The main man-introduced perturbation being the injected fluid, at least the physical mechanism of pore pressure generation should be included in the

approach to do so. This was out of scope here, due to a lack of available public data.

Another point worth considering is the effect of the background stress. Some studies have argued that most of the seismicity occurred according to it (Cesca et al., 2014; Salo, 2016), especially during the strongest phase of the sequence (see Fig. 2). The current knowledge in its regard deriving from regional studies, detailed site measurements would add robustness to any analysis and is therefore advised for upcoming studies in the area. Moreover, it seems logical to think that seismicity was

triggered due to various interacting mechanisms; among these, static stress triggering was most likely significant. While theoretical knowledge of all of them in the context of anthropogenic seismicity is currently well established (e.g. Ellsworth, 2013; TNO, 2014), the interaction of more than 2 of them is rarely considered in published studies based on numerical approaches. Here, apart from pore fluid generation, poroelastic stress and mass changes could also be involved if a larger time scale is considered, other than the instantaneous variation reflected in our modelling (note that the CASTOR UGS was

placed in the depleted Amposta oil field, which had formerly been exploited).

Finally, the authors note that in the applied constant apparent friction model, the pore fluid pressure effect is treated by means of $\mu'$. Using other models could lead to dissimilarities in the results under certain conditions (see Beeler et al., 2000).

## 6. Conclusions

Our results point static stress transfer to have been a destabilizing mechanism, as shown by positive $\Delta CS$ in the nodal planes

of 7 out of 8 events of the studied sequence; nevertheless, we note that combined effects with other triggers could have had their saying as well. Hence, we reflect on stress transfer by neighbouring earthquakes as a (partial) destabilizing factor in the series. Known faults' study establishes the East 4 structure as the one with higher probability of having been activated during the series. Further constraining this statement is difficult because of the hypocentral location uncertainty. Consistent with our findings, the seismic cycle concerning the Main Fault's characteristic earthquake (around $M_w$ 6.0) was not shortened by the

experienced events; smaller events' occurrences could have been hastened to some extent.

This contribution is the result of a rigorous analysis based on currently available information. However, further resources (e.g. injection and well pressures data, providing essential information to manage changes in the effective stress state) would have allowed a different approach which cannot be accounted for. Recalling this fact is important as case complexity advocates for future studies to be provided with all injection-related data so as to make more insightful statements.



**Figure 1: Location of the study; the Castor Platform is represented with an orange star. a) General view showing existing permanent seismic stations around the UGS and main stations in Catalonia area. ECOL station was set up after the beginning of seismicity and was not used in the majority of the available earthquake locations. See legend for agencies. b) Sketch of the UGS structure after the Shell seismic profile in Seeman et al. (1990), vertical scale is in two-way traveltime. A rough approximation of the reservoir body is depicted in yellow, and stratigraphic markers are shown in blue and green (modified from Cesca et al., 2014). c) Detailed view of known faults in the area modified from Cesca et al. (2014), according to the red area highlighted in a).The map view is plotted at the approximate depth indicated by the discontinuous line in b).**





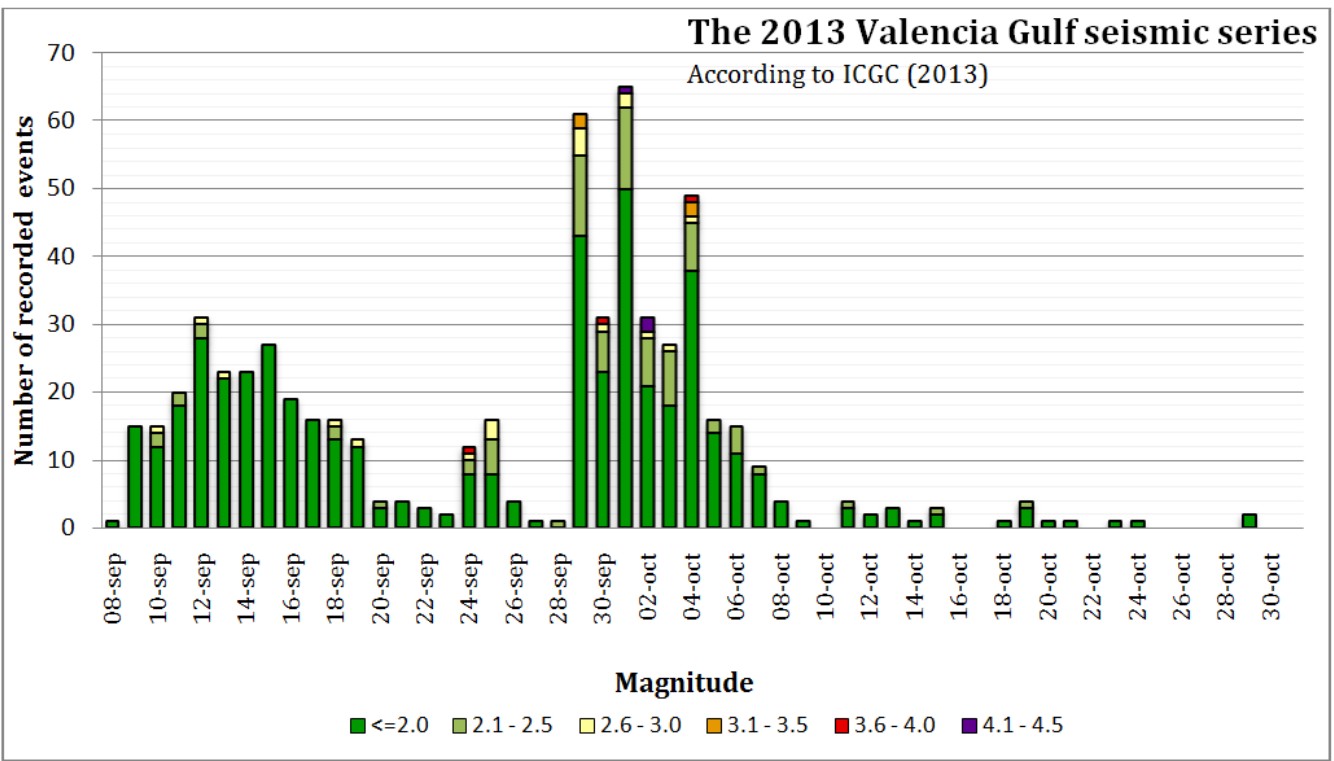

**Figure 2: Histogram of Castor's seismic sequence, showing the number of earthquakes per day and their magnitudes. Two separate phases can be distinguished in this sequence. The first one would last until September 19th, just two days after injections were stopped, and maximum magnitudes did not surpass M 3. After four days of almost no seismicity, the first felt earthquake took place on September 24th (M 3.7), and high levels of seismicity with three M 4 earthquakes were recorded during the two following weeks. Modified from ICGC (2013).**

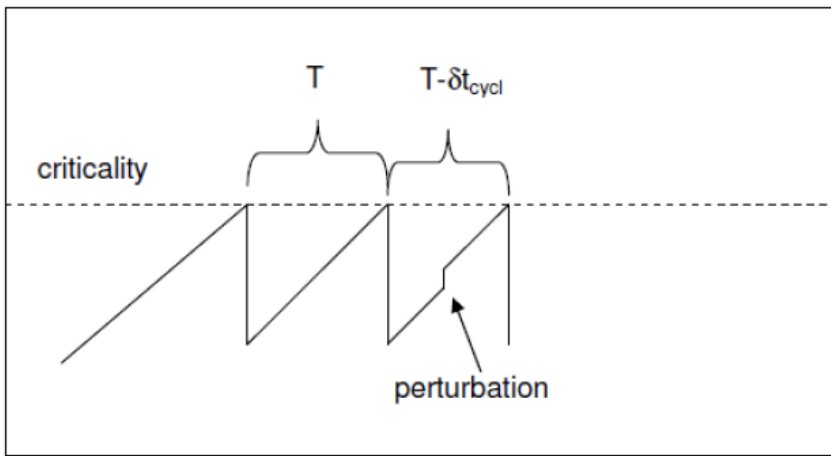

**Figure 3: Representation of the shortening of the seismic cycle due to a perturbation which could be man caused. In this case the perturbation is of positive nature, but could be of negative nature as well. The latter is obviously not concerning. Modified from Baisch et al. (2009).**



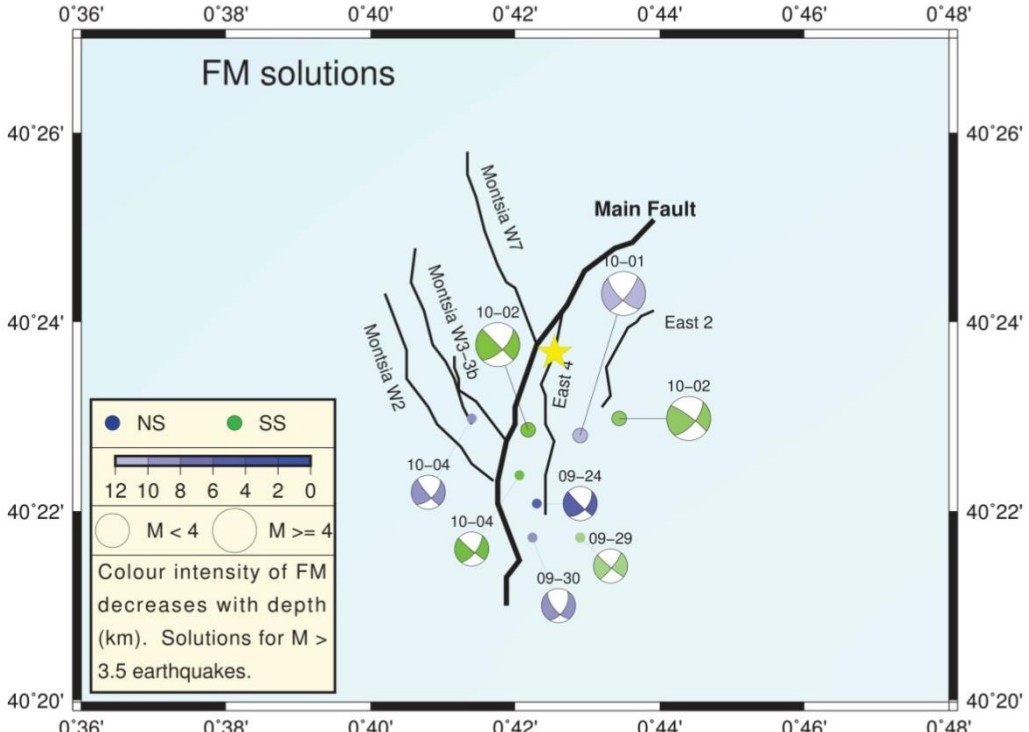

**Figure 4: Obtained solutions for the 8 main events. Dots indicate the location of each *beachball* plot. Mapped faults are indicated as well. In the legend, NS refers to those mechanisms with normal component, whereas the others (SS) are essentially pure strike slips.**




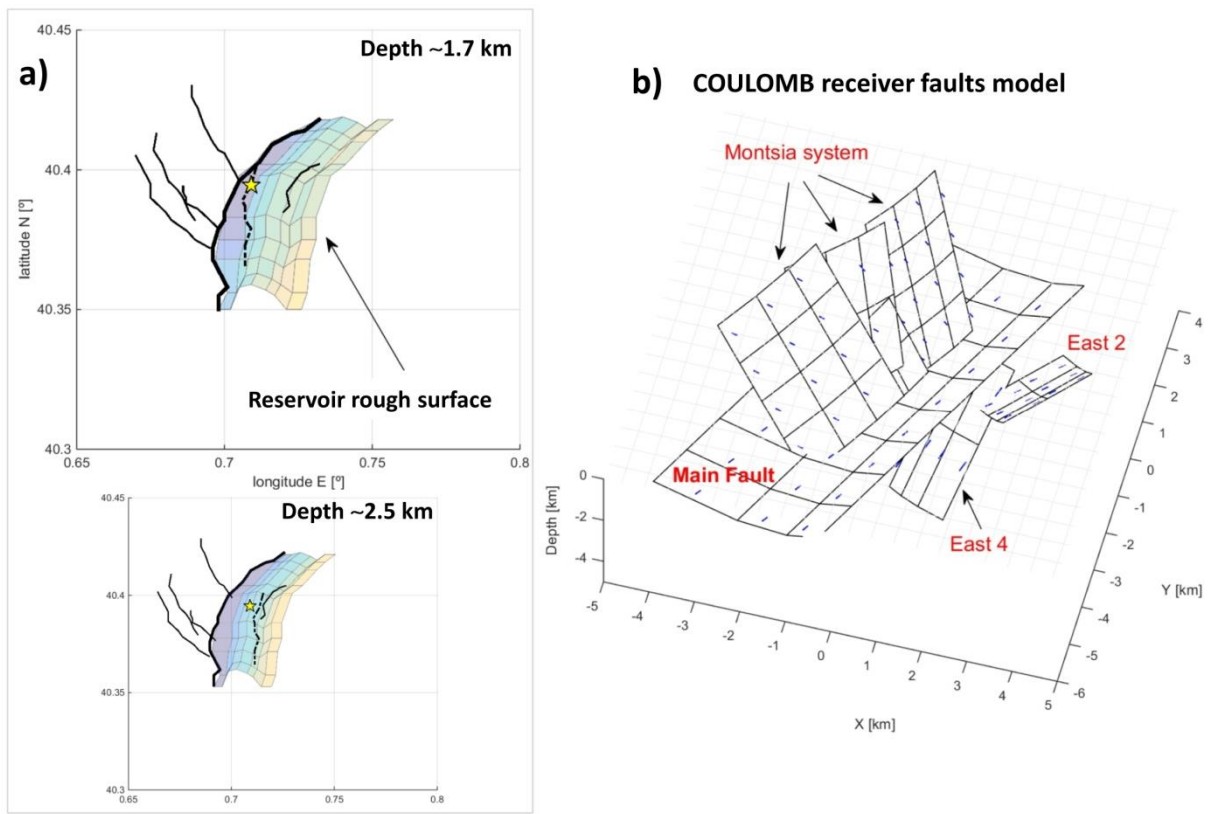

**Figure 5: a) Horizontal slices of the geometrical 3D model of known faults, near reservoir top and base depth, derived from references indicated in the main text. Fault traces are shown as black lines, continuous if dipping to the west and discontinuous otherwise (East 4). b) Receiver fault model used in COULOMB calculations, generated using a) as an input. The Montsia system and the Main Fault reach depths near surface, while the East faults' shallowest point is about reservoir top depth (compare Fig. 1b).**





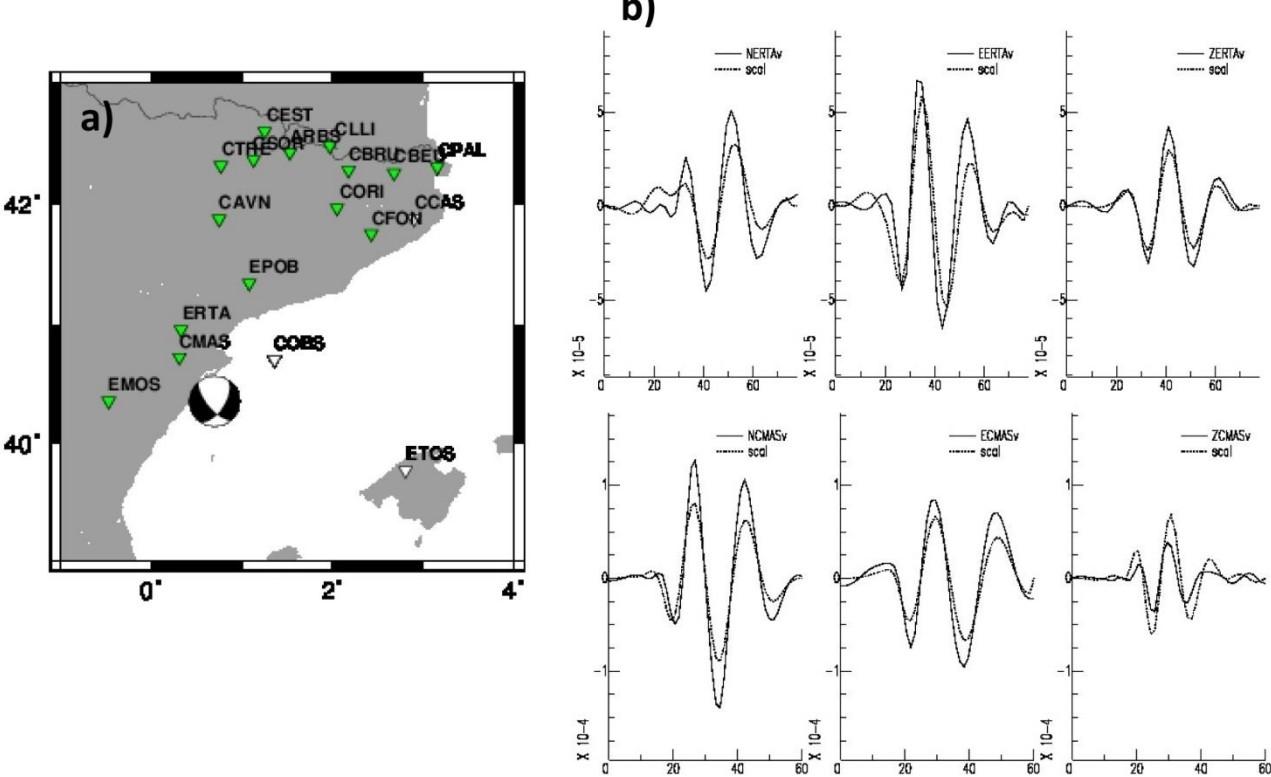

**Figure 6: a) output map with the FM solution for the Mw 3.9 occurred on September 30th, 2013, its location, and the stations used to compute it (green triangles). b) Waveform fit at stations ERTA and CMAS for the named event. The recorded waveforms are plotted with a continuous line, the discontinuous being the adjusted one.**

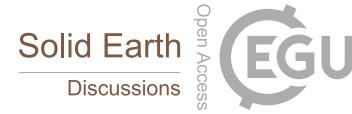



**Figure 7: Coulomb stress resolved on the nodal planes. Each subplot shows the FM solution with ΔCS due to the nodal plane of the previous earthquake(s) slipping, on both the nodal plane(s) of the old and the upcoming event. Past nodal planes are shown as red rectangles (faults with slip). Note that colorbar scale saturation changes as sequence evolves.**



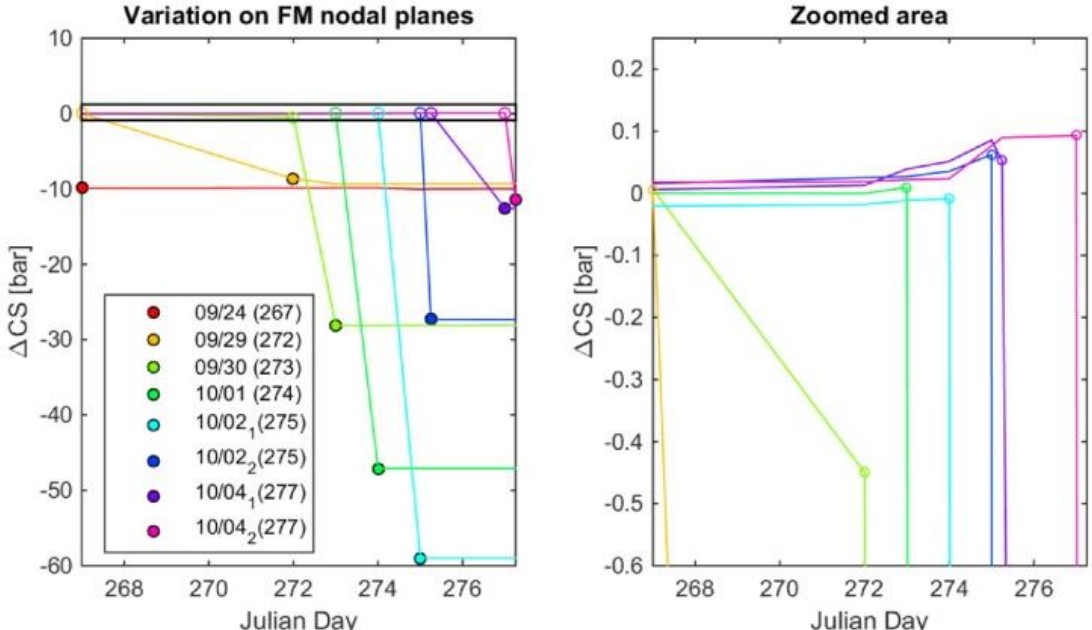

**Figure 8: Time series of ΔCS on the source fault planes. Each color refers to one different plane, with empty circle markers being the state before the quake and filled markers afterwards. The black box in the left subplot limits the zoomed area on the right.**

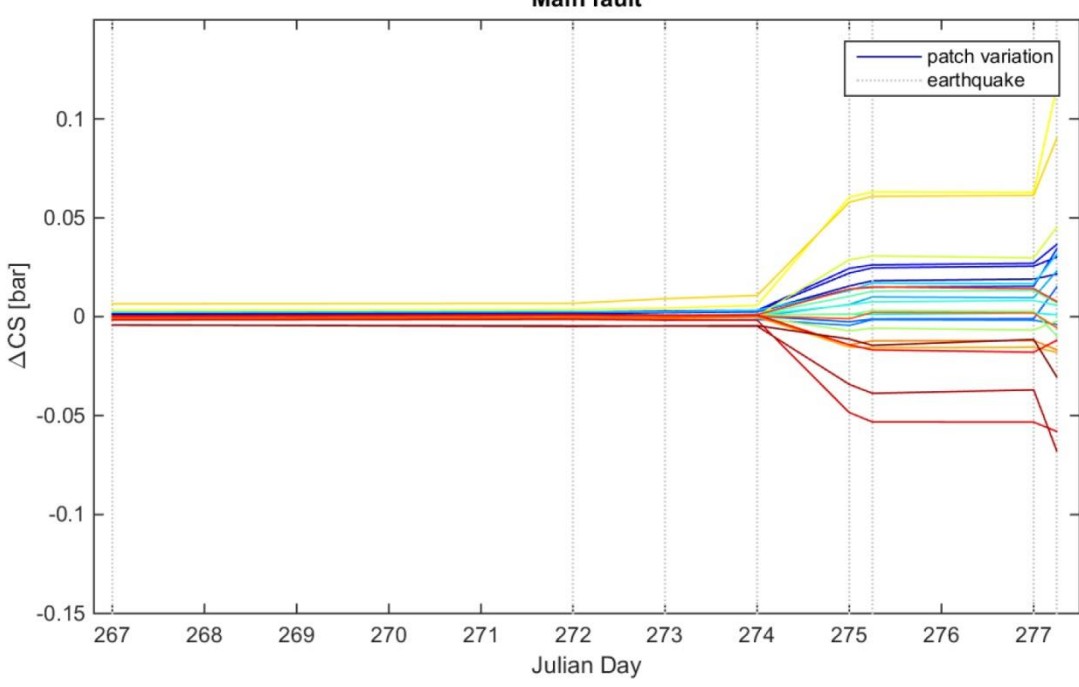

5  **Figure 9: Time series of ΔCS computed onto each patch of the Main Fault. The discontinuous vertical lines indicate the occurrence of an earthquake (note that days 275 and 277 have two occurrences). Colors are used to better distinguish each patch, and do not accord with ΔCS being positive or negative (which is represented in the vertical axis scale).**





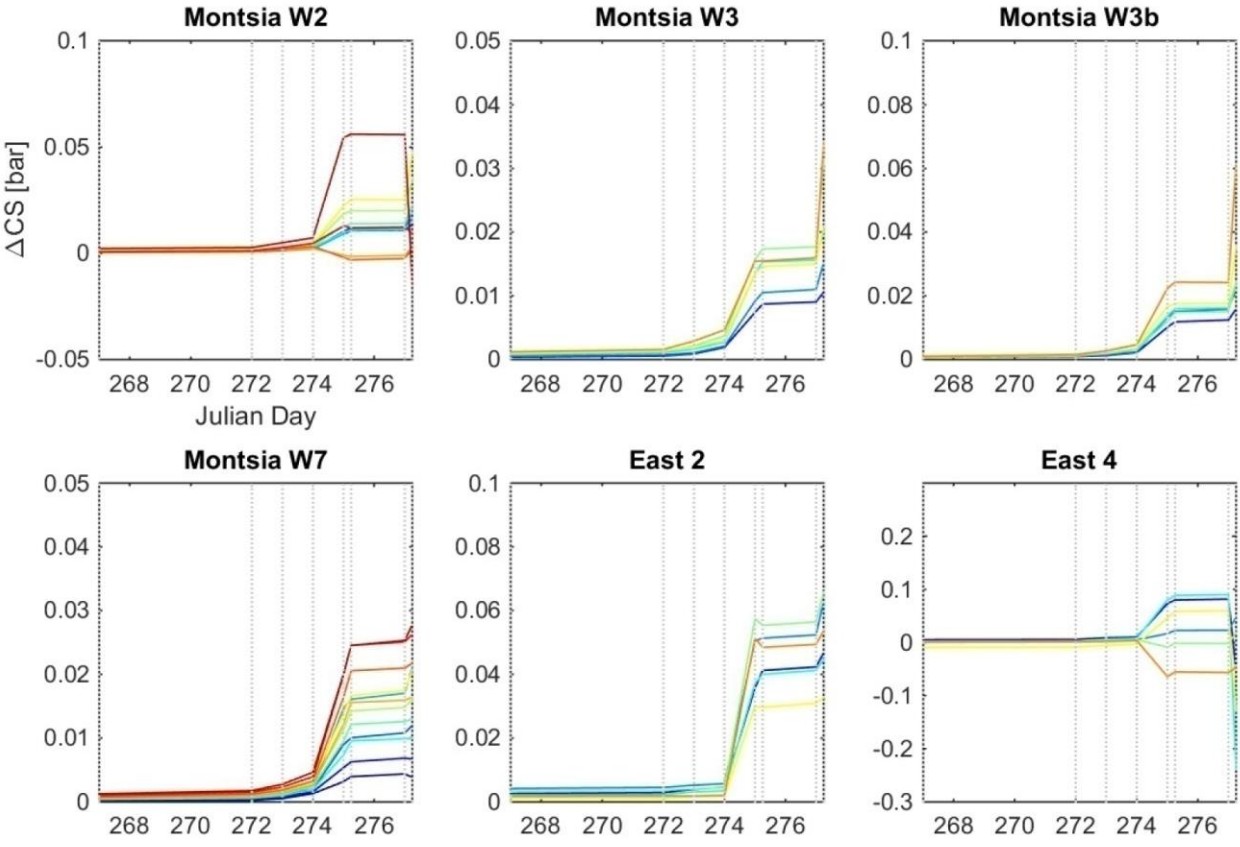

**Figure 10: Time series of ΔCS computed onto each patch of the indicated receiver faults. The discontinuous vertical lines indicate the occurrence of an earthquake (note that day 267 has one occurrence, and 275 and 277 have two). Colors are used to better distinguish each patch, and do not accord with ΔCS being positive or negative (which is represented in the vertical axis scale). Vertical scale changes among panels.**




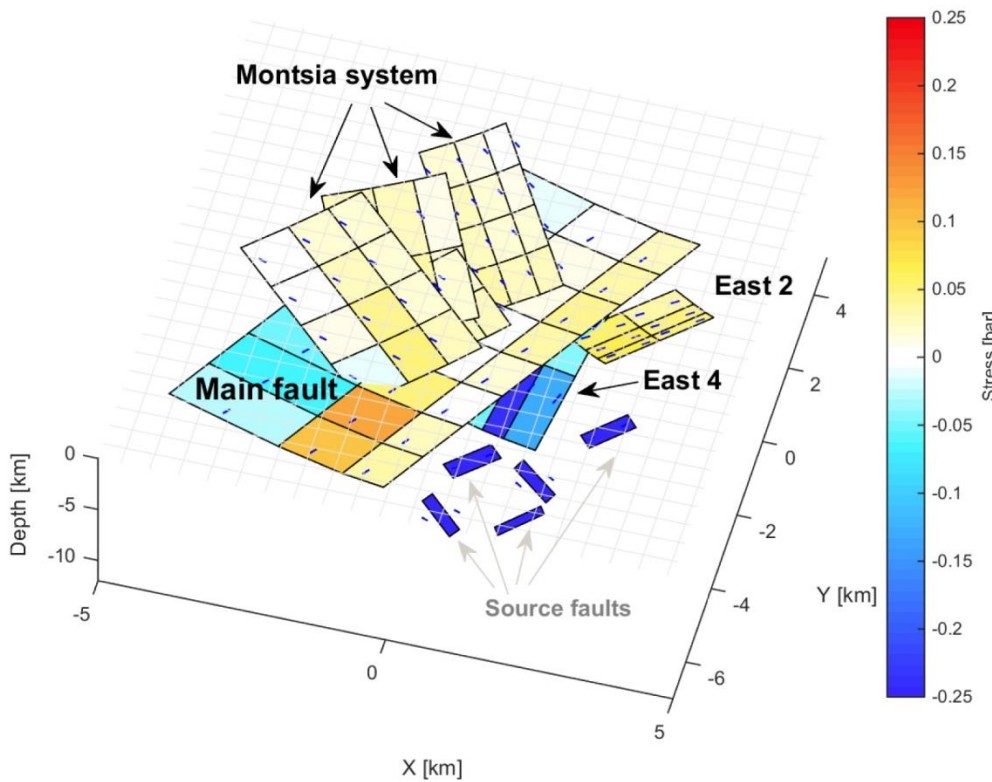

**Figure 11: Final stress state after the 8 studied events. Smaller rectangular patches which are deeper correspond to the source faults according to FM information. The orientation of the plot view is chosen so that all receiver (mapped) faults are viewed at the same time.**





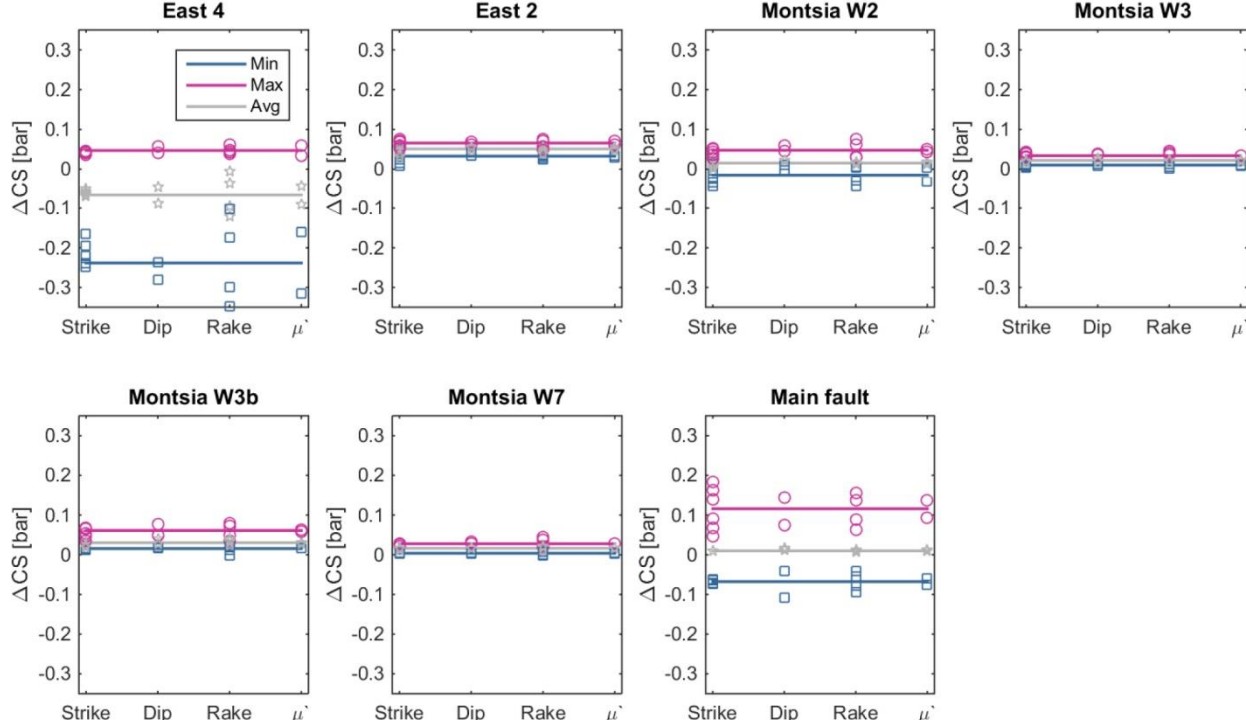

**Figure 12: Final stress state variation on each receiver fault, depending on which parameter is changed. Continuous horizontal lines indicate the result when the assumed best estimation for each parameter is taken. Minimum and maximum refer to the peak values achieved on a patch of the fault, while the average is the mean value computed over all the patches. Solid markers (circles, stars and squares), refer to variations with respect to the best estimation (according to parameter variation in horizontal axis and color).**



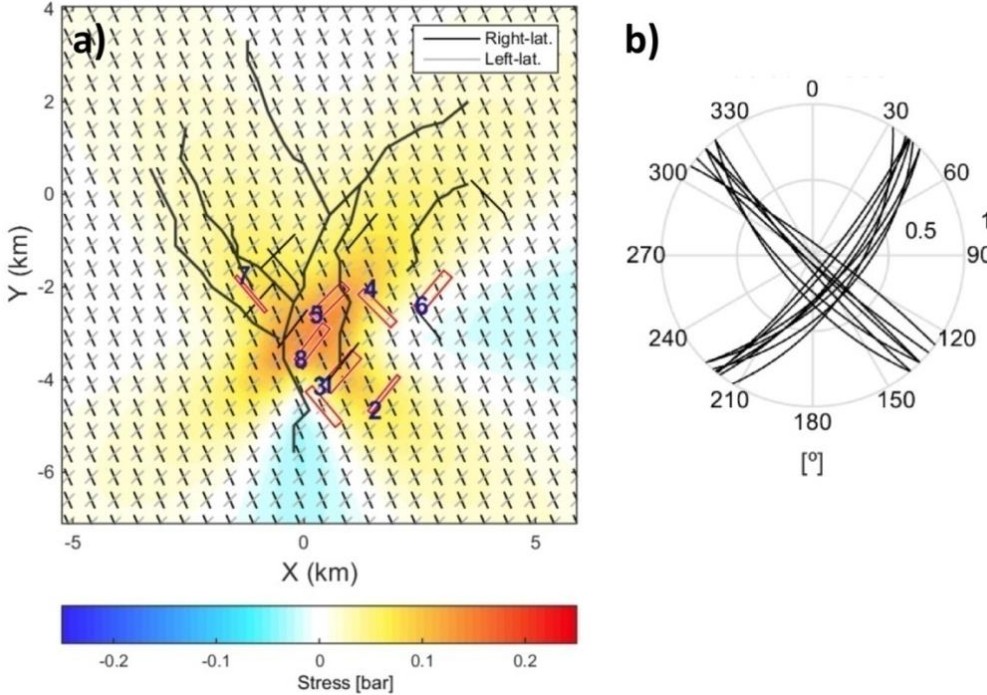

**Figure 13: a) OOFP study. Red rectangles represent source faults, and the receiver faults' traces are shown as well. Right lateral and left-lateral optimal orientations according to assumed background stress are shown as well (see legend). Note that most of the receiver faults are well orientated with respect to the background stress (close strike directions to those of OOFP). The stress map view is plotted at 1.7 km of depth. b) Stereographic projections of both families derived from FM solutions (NW-SE and NNE-SSW to NE-SW), represented on a polar sphere.**

**Table 1: Shortening of the seismic cycle of a $M_w$ 6.0 event on the Main Fault. The result is given in % for 4 values of Ds ($\Delta\sigma_{ss}$, $\Delta\sigma_{ds}$ derived from calculation as shown in Eq. 6, and $\Delta\sigma_{max}$, $\Delta\sigma_{min}$ being the assumed range). Once the $T_r$ is estimated, the worst assumption is taken to give the result in years (assumed stress drop $\Delta\sigma_{min}$). Of all variations in strike, dip, rake, µ' and depth, the case in which the computed $\Delta CS$ was higher is selected to provide a conservative approach. See Appendix C for best estimation of parameters.**

| $\delta_{cyc}$ - MainFault | | $M_w = 6.0$ | | | | |
|---|---|---|---|---|---|---|
| | $\Delta CS_{mean}$ [bar] | $\delta_{cyc}$ [%] | | | | $\delta_{cyc}$ [y] |
| Parameter | | $\Delta\sigma_{ss}$ | $\Delta\sigma_{ds}$ | $\Delta\sigma_{min}$ | $\Delta\sigma_{max}$ | $T_r = 5191$ y (SD = 6310) |
| **Best** | **0.010** | **0.025** | **0.019** | **0.100** | **0.033** | **5.200** |
| Source strike | 0.009 | 0.022 | 0.017 | 0.089 | 0.030 | 4.615 |
| Source dip | 0.017 | 0.044 | 0.033 | 0.174 | 0.058 | 9.025 |
| Source rake | 0.011 | 0.029 | 0.021 | 0.113 | 0.038 | 5.892 |
| µ' = 0.6 | 0.011 | 0.029 | 0.021 | 0.114 | 0.038 | 5.911 |
| Depth | 0.044 | 0.110 | 0.083 | 0.438 | 0.146 | 22.726 |





**Table 2: Shortening of the seismic cycle of a M$_w$ 5.0 and a M$_w$ 4.5 event on the Main Fault. Indications as in Table 1.**

| δ$_{cyc}$ - MainFault | | M$_w$ = 5.0 | | | M$_w$ = 4.5 | | |
|---|---|---|---|---|---|---|---|
| | ΔCS$_{max}$ [bar] | δ$_{cyc}$ [%] | | δcyc [y] | δ$_{cyc}$ [%] | | δcyc [y] |
| Parameter | | Δσ$_{ss}$ | Δσ$_{ds}$ | Tr = 1303 y (SD = 1584) | Δσ$_{ss}$ | Δσ$_{ds}$ | Tr = 411 y (SD = 910) |
| **Best** | **0.116** | **0.958** | **0.720** | **12.487** | **0.927** | **0.698** | **3.816** |
| Source strike | 0.182 | 1.507 | 1.132 | 19.637 | 1.458 | 1.098 | 6.001 |
| Source dip | 0.145 | 1.195 | 0.898 | 15.581 | 1.157 | 0.871 | 4.761 |
| Source rake | 0.156 | 1.290 | 0.969 | 16.810 | 1.249 | 0.940 | 5.137 |
| μ' = 0.6 | 0.138 | 1.137 | 0.855 | 14.819 | 1.101 | 0.829 | 4.529 |
| Depth | 0.640 | 5.287 | 3.974 | 68.912 | 5.118 | 3.854 | 21.058 |

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

## 8. Appendices

### 8.1 Appendix A: Velocity model

The velocity model used for calculations is shown in Table A.1.

**Table A1: FMNEAR webservice speed model. $Q_P$, $Q_s$ are the seismic attenuation coefficients (Higher Q means lower attenuation).**

| Crust Velocity model | | | | | | |
|---|---|---|---|---|---|---|
| Layer | Layer base depth [km] | $V_p$ [km/s] | $V_s$ [km/s] | Density [kg/cm$^3$] | $Q_P$ | $Q_s$ |
| 1 | 0.6 | 3.3 | 1.75 | 2 | 200 | 100 |
| 2 | 1.4 | 4.5 | 2.6 | 2.3 | 350 | 175 |
| 3 | 3 | 5.5 | 3.18 | 2.5 | 500 | 250 |
| 4 | 25 | 6.5 | 3.75 | 2.9 | 600 | 300 |
| Mantle | - | 8.1 | 4.68 | 3.3 | 1000 | 500 |



## 8.2 Appendix B: FM solutions

**Table B1: FMs solutions summary table .Sorted by input magnitude (highest to lowest). Solutions obtained by Frontera et al. (2013), IGN (2013) and Cesca et al. (2014) are also shown to ease comparison. Frontera's et al. (2013) solutions were obtained using the same method (FMNEAR) and model (default).**

| FM SOLUTION TABLE | | | | | | | | | | | | | | | | |
| Earthquake location data (ICGC, 2015) | | | | | FM computation and result | | | | | | | | | | Previous studies | | |
| Date – time UTC | Lat. [°] N | Lon. [°] E | Depth [km] | $M_L$ | Input depth [km] | FM depth [km] | Strike [°] | Dip [°] | Rake [°] | RMS | Conf. Index [%] | Comp. used | $M_W$ | Beach ball plot | Frontera et al. (2013) | IGN (web catalog) | Cesca et al. (2014) |
|---|---|---|---|---|---|---|---|---|---|---|---|---|---|---|---|---|---|
| 2013/10/02 23:06:50 | 40.381 | 0.703 | 5.9 | 4.3 | 3 | 5 | 135.0 | 90.0 | -154.8 | 0.572 | 87 | 31 | 4.2 | | | | |
| 2013/10/0103 :32:45 | 40.38 | 0.715 | 5 | 4.2 | 8 | 11 | 135.0 | 70.0 | -164.6 | 0.5 | 88 | 24 | 4.1 | | | | |
| 2013/10/02 23:29:29 | 40.383 | 0.724 | 0.9 | 4.1 | 5 | 6 | 40.0 | 65.0 | 8.8 | 0.6 | 74 | 26.0 | 4.0 | | | | |
| 2013/09/30 02:21:17 | 40.362 | 0.704 | 3.2 | 3.9 | 4 | 8 | 140.0 | 65.0 | -142.5 | 0.6 | 83 | 32 | 3.9 | | | | |
| 2013/10/04 08:49:48 | 40.383 | 0.69 | 8.9 | 3.8 | 5 | 8 | 45.0 | 60.0 | -9.5 | 0.5 | 82 | 18.0 | 3.7 | | | | |
| 2013/09/24 00:21:50 | 40.368 | 0.705 | 4.9 | 3.6 | 4 | 5 | 135.0 | 85.0 | -144.4 | 0.539 | 83 | 13 | 3.6 | | | Not available | |
| 2013/10/04 09:55:20 | 40.373 | 0.701 | 5 | 3.5 | 3 | 3 | 130.0 | 85.0 | -157.6 | 0.5 | 79 | 10.0 | 3.6 | | | | |
| 2013/09/29 22:15:48 | 40.362 | 0.715 | 0 | 3.5 | 5 | 8 | 130.0 | 85.0 | -169.9 | 0.6 | 84 | 28.0 | 3.6 | | | | |



### 8.3 Appendix C: Calculation parameters

The following table provides calculation parameter values and ranges used. The method used to determine each and/or references used are shown as well. Best estimation parameters (those used in the main text plots unless indicated) are: μ' = 0.4, source faults strike, dip and race according to FM solutions, receiver faults strike and dip according to previous mappings and rake assumed as the average value of that of the corresponding nodal plane family derived from FM solutions.

**Table C1: Parameter selection and variations in order to compute Coulomb stress changes and shortening of the seismic cycle. Regional stress is only used in Fig. 13.**

| Parameter | Unit | Values | Source &\| Method |
|---|---|---|---|
| Young's modulus ($E$) | Pa | 8E+10 | Assumed value (Toda et al.,2011b) |
| Poisson ratio ($\nu$) | - | 0.25 | Assumed value (Toda et al., 2011b) |
| Effective friction ($\mu`$) | - | 0.2, 0.4, 0.6 | a) From $\mu$, $\gamma_r$ and stress regime (Zoback, 2007). b) Recommended values (King et al., 1994; Hardebeck et al., 1998; Sumy et al., 2014) |
| Regional stress regime | - | Strike slip | FM solutions (this study; Frontera et al., 2013; IGN, 2013; Cesca et al., 2014) |
| Regional stress orientation | ° | $S_H = 23 \pm 9$ $S_h = Sh + 90$ $S_v$ = vertical | Schindler et al. (1998), Heidbach et al. (2008), Cesca et al. (2014) |
| Regional stress magnitude (z = 1.7 km) | Bar | $S_1$: 411-391 $S_2$: 391-371 $S_3$: 248-241 | Critically stressed crust and frictional equations (Jaeger and Cook, 1979; Zoback, 2007). Water table negligible. |
| Source faults: strike, dip, rake, geometry and net slip | °, m | Strike: FM ± 10° Dip: FM ± 10° Rake: FM ± 20° Depth: FM - 3 km | Strike, dip and rake from FM solutions in this study. Rectangular geometry (L=1km = 2w). Net slip to accord with Magnitude of the event, for a given geometry (e.g. Aki and Richards, 1980) |
| Receiver faults: strike, dip, rake and geometry | °, m | Strike: References Dip: References value ± 10° Rake: FM avg. value ± 20° | References in the text. Rake to accord with FM-derived stress regime. |
| Main fault rupture area ($RA$) | Km$^2$ | 75-2.75 | From dimensions in the derived model, accounting for curvature (max value). Wells & Coppersmith (1994) otherwise. |
| Main fault Moment magnitude ($M_w$) | - | 6.0-4.5 | From RA (Wells & Coppersmith, 1994) |
| Shear modulus ($G$) | Pa | 3.2E+10 | According to $E$, $n$ |
| Main fault slip rate ($SR$) | mm/y | 0.04-0.63 | Perea (2006), Garcia-Mayordomo et al. (2015), |
| Stress drop ($\Delta\sigma$) | Bar | $\Delta\sigma_{ds}$, $\Delta\sigma_{ss}$ calculation derived. $\Delta\sigma_{max}$, $\Delta\sigma_{min}$ 30 and 10 respectively. | Calculated for strike and dip slip (Lay and Wallace, 1995). Likely range (Baisch et al., 2009) considered as well. |

### 9. Acknowledgements

We would like to thank the Institut Cartografic i Geologic de Catalunya (ICGC) and the Universitat Politecnica de Catalunya (UPC), collaborating institutions whose funding allowed the completion of this study. The authors thank Prof. William



Ellsworth and Dr. Martin Schoenball as well, for their remarks on how to properly address a case of induced seismicity, and on computing Coulomb Stress Changes respectively.

**10. Competing interests**

The authors declare that they have no conflict of interest.

