# Peer review of "Earthquake static stress transfer in the 2013 Valencia Gulf (Spain) seismic sequence"

_Solid Earth, 2016_

## Referee Comment (RC1) · Anonymous Referee #1 · 5 Jan 2017

**Earthquake static stress transfer in the 2013 Valencia Gulf (Spain) seismic sequence**
*by L. Salo et al.*

*Submitted to Solid Earth*

**General comments**

In its current form the draft reads not very well and its major points are not clear and convincing enough. It needs a major revision to be worth publication. First of all, I think that this manuscript could significatively benefit from an overall restyling. I recommend a general and thourough improvement of the English (see *Technical issues*) and a more organized, concise and focused presentation of the method and results. The authors should be clerer in presenting and justifying their initial assumptions and stress out their goals (which is the final message and why is it significant in this context?). In this study the authors should bring into focus with more enphasis the specific analysis and result interpretation, which represents the innovative part of the study. Their results should be significative and presented in a convincing way. The methodology and the numerical code used for the static stress estimatimation are indeed already quite well known in the literature and do not represent novel tools.

The authors are studying the role of static stress redistribution due to earthquake mutual interactions during a seismic sequence that is probably induced by the injection of cushion gas for gas storage. In their analysis they are ignoring any influence that could come (and would probably be significant) from the triggering effect of pressure changes caused by the injection. This is a limiting but possible assumption if the focus of the study is on the role of earthquake interactions only (e.g. *Baisch et al.* 2009; *Schoenball et al.* 2012; *Catalli et al.* 2013). However, the authors have to stress out more clearly their specific goal and strong assumption from the beginning and touch the general issue of induced seismicity to give an overview on all the possible triggering mechanisms involved in such a case. The feeling after reading the text is that they comment their assumption of ignoring any other triggering effect during the final sections Discussions and Conclusions and it sounds like the study is incomplete due to shortage of data more than having a different initial aim.

**Specific comments**

I think that in this study there is an important confusing point: the calculation of ΔCFS made both on the *source fault planes* (section 4.1) and on the *mapped fault planes* (section 4.2). Is this distinction worthy and why? I would avoid to say *source faults* when they are used thereafter also as receivers. Additionally, it is not clear which are the source planes used for estimating ΔCFS on the mapped faults; are also the mapped faults contributing to the cumulative ΔCFS as sources? The authors should be clearer in explaining which are the sources/receivers for each different case they present and list all relative information, as for example in Table B1. It is very difficult to interpret Figure 11 with current information; in particular, one does not easily know the time evolution of ruptures relative to the faults or fault's patches. Did for example the East 4 fault slipped at the beginning or in the middle of the sequence, or maybe at the end? Which are the sources contributing to the cumulative ΔCFS estimated on top of the East 4 fault? How does ΔCFS on this specific fault evolve with time? This would help undertsanding its negative ΔCFS.

Another crucial problem regards the interpretation of results in terms of ΔCFS: the authors conclude that the fault named East 4, which shows a negative cumulative ΔCFS at the end of the sequence (Figure 11), is therefore the most likely to have slipped. However, another

possible interpretation could be that the influence in terms of cumulative ΔCFS of all the previous events on the fault East 4 is negative, i.e. the fault is not favored to slip by the cumulative stress redistribution. The stress drop on the fault might indeed have been caused also by its own slipping phase, but for a better understanding of this point one needs to know the fault's stress state before and after the time of its slipping in relation with the other events in the sequence (time evolution). In other words, ΔCFS on the East 4 fault is negative before or after its own rupture? This is also not clear from Figure 10. All these aspects should be clarified for a proper interpretation of results. Moreover, the fault is only partially visible in Figure 11.

Referring to Figure 12, how are the best estimations of the parameters caculated/assumed? Why do the authors limit the analysis to just four parameters (strike, dip, rake and apparent friction)? Why do the authors perform this analysis referring to the mapped fault and not also the the so called source faults? The table that describe free parameter variations (now Table C1, which describes all parameters involved in the methodology) should be a dedicated table for the sensitivity analysis. Why do the authors vary the strike of the sources and not of the receivers? I think both have to be randomly perturbed within 20 degrees for understanding the sensitivity to the strike angle (the same for dip and rake). They might reproduce this way a large number of realizations for a more probabilistic based sensitivity analysis. The same concept would apply also for the nodal plane, depth and friction coefficient.

Can the authors explain why *the fact that the mean value in the East 4 fault is near -0.1 bar, supports the idea of this structure to have slipped* (section 4.3.1 lines 22-24)?

The final message about static stress transfer as triggering mechanism is not enough convincing (section 5.1): what is the take home message? The fact that the static stress acted only as a partial triggering factor is not a strong finding itself. The conclusions are dispersive.

The three first lines of Conclusions are confusing: for the East 4 fault a negative ΔCFS meant possible slip (see before), while here a positive ΔCFS resolved onto 7 of the 8 FM means possible destabilization. It looks like the authors are changing their point of view of the same process. They should be clearer and more consequential.

**Technical issues**
Some examples of intricate, hard to understand sentences: p.2, lines 16-16; 22-23; 27-29 (here the authors seem to justify the fact that they use FM solutions as sources of Coulomb stress change. Why do they need it? This is a very common assumption in the literature); p.3, lines 18-19 (explain which error is minimized and how). P.4, lines 4-5: reduction of what? try to be clearer; lines 11-13: explain better this concept of conservative or non-conservative assumption and which of the two you are following. P. 5, lines 3-4.

Section 2 is probably too long and confused. I would suggest to respect the order resources/available data - methodology description – Coulomb model assumptions. I suggest to create a section on the methodology (Coulomb and seismic cycle) and one on the model assumptions (sources and receivers). The section about uncertainties should be independent and self-consistent. I think that the factors of uncertainty are several and regard both the methodologies involved in this study (the Coulomb stress estimation and the impact on the recurrence time). The authors should list them clearly and explain why do they analyse only some of them and how.

Section 2.1: the authors allude to *various criteria to determine failure conditions* and then they

do not mention any of them. In general, if one refers to an issue, then one should at least spend a few words on that or omit it at all. Equation 1: why do the authors use the primes for $c$ and $\sigma$? The simpler notation, the better. Later in equation 3 the prime disappears for $\Delta\sigma$ and appears (correctly) for $\mu$ (but this causes confusion for the reader). $\Delta CFF$ is defined but $\Delta CS$ (line 7) is not. More generally on this section, it looks like that the authors are jumping from a formula to another for describing the concept of Coulomb failure criterion but they forget to give to the reader some simple elements for understanding a quite intuitive concept. If they want to start from the Mohr-Coulomb theory, they could then show a figure with a general description of the Mohr-Coulomb diagram. This would also be a good expedient for describing the possible effect (even though neglected in this study) of a pressure increase.

The authors should be more careful in explaining the precise conditions under which equation 4 is valid, i.e. the pressure effect is taken into account only under the undrained condition (no fluid flow is considered) and the solely contribution to the pressure is given by the compressional terms of the stress tensor. They should remind that the pressure change due to the injection is still completely neglected in equations 3 and 4 and in general in their study.

Why do the authors think that working with the software Coulomb 3.3 allows to deal with tridimensional complexity and not with other numerical models of the Coulomb stress redistribution? How do they justify their choice of calculating the stress change at 11 km depth (line 22, p.3)?  From table B1 one can see that the 8 major events used as sources/receivers are all shallower than 5.9 km with an ecception of an event at 8.9 km. Additionally, Figure 5b does not help making a clear picture of the depths of the receiver faults. The depth of calculation of $\Delta CFS$ plays a significative role and should be discussed with greater attention.

Section 2.2: I would add some more explanations and comments referring to equations 5 and 6. Both equations imply the assumption of the concept of the characteristic earthquake. This is a strong assumption and it is worth some more discussion. Equation 5 implies that the average stress drop is released from a characterstic earthqake only, so that $\Delta\sigma/T_r$ corresponds with the regional stressing rate; all other possible phenomena of stress release are ignored. Equation 6 on the other hand implies the existence of a characteristic length for the source of a characteristic earthquake. I think that the authors should briefly comment on these hypotheses (why do they think these hypotheses are realistic in their context? What are the limitations?)

In general, I find that sometime in this draft the references to other studies for justifying or explaining some assumptions are ambigous. For example, equation 5, p. 4: neither in *Harris* (2000) nor in *Baysch* et al. (2009) I can explicitly find any reference to this equation but solely hints to the idea that a stress change can affect the recurrence time of large earthquakes. I would find a reference as for example to the study of *Parsons* (2005) and references therein much more focused on the problem.

The *Main Fault* cited for the first time in correspondance of equation 9 and then in Figure 9 should be clearly defined. Please also define *SR*.

Section 2.3.1 can be merged with Data and Resources.

Section 2.3.1: rather than giving information on details about the input/output data of the method FMNEAR used for FM estimations, I would focus on the approach itself and the reliability of the method. CFS estimations are indeed very sensitive to the FM solutions used for sources and receivers and their relative hypocentral distance. Why do the authors decided to use FM solutions obtained via FMNEAR? Why did they not use solutions already published in the cited literature (*Frontera et al.* (2013); *ING* (2013) and *Cesca et al.* (2014)? How sensisive are

their results to the different solutions?

Section 2.3.2: in the first paragraph of this section the English and wording needs a substantial improvement. The method for selecting the nodal plane is unclear. However, I do not agree/do not understand the idea that the nodal plane is selected referring to the highest ΔCFS. The authors need to justify their assumption. An alternative idea could be to quantify the difference when using a different nodal plane solution for each event or by using random extrapolated nodal plane solutions for a more probabilistic approach. P.5 from line 32: are the authors explaining how do they estimate the slip of each event? Does it come from the magnitude? It is unclear.

Section 2.3.3: what I would need to understand clearly here is on which receiver planes do the authors estimate ΔCFS and why. From this paragraph one thinks that the FM solutions are used as sources and the geological faults as receivers. However, already in the abstract the authors say that :"…*the evolution of static stress is quantified both on fault planes derived from focal mechanism solutions…and on the previously mapped structures…*". The issue is touched then again in section 4. I think that the authors need to be more concise and precise on this assumption from the beginning and explain the reason of their assumptions.

Section 2.4: this section is not complete. A reader needs to understand which are the parameters at play; which of these parameters influence uncertainties the most; how do the authors perform a comprehensive sensitivity analysis and major findings. To be more concise, tha authors may merge section 2.4 directly into section 4.3.

Section 4.4: first line: the poor reader needs to check the two tables ahead and equation 5 behind to understand what you are discussing about. The tables can be merged together. The case *Best* should be better defined and justified. The tables are not easily comprehensible. Where are *the results for the best estimate* plotted? There is no reference to a Figure.

Section 5.2, the first lines are uncomprehensible.

In Figure 1 would be interesting to see also the injection rate.

Figure 4 can be merged into panel c of Figure 1.

Figures 5 and 11 can be merged together into an unique Figure.

Figure 7 is not clear and one cannot follow the relative comments reported in section 4.1. Projecting ΔCFS on the FMs can just give a rough idea of the stress change (positive/negative) and information on the spatial distribution is lost. Why do the authors decided for this kind of presentation? Which is the message they want to give through this figure?

Figure 8: which are the events causing a ΔCFS on the FM of the event occurred the 09/24?

Figure 11: report in the figure, close to the corresponding patch, the identifying letter+number already reported in Figure 10.

Tables B1 and C1 are fundamental for the interpretation of the methodology and results, why are there in the appendix?

**References**

Baisch, S., et al. (2009), Deep heat mining Basel—Seismic risk analysis, Tech. Rep., Serianex.

Catalli, F., M.-A. Meier, and S. Wiemer (2013), The role of Coulomb stress changes for injection-induced seismicity: The Basel enhanced geothermal system, Geophys. Res. Lett., 40, 72–77, doi:10.1029/2012GL054147.

Parsons, T. (2005), Significance of stress transfer in time-dependent earthquake probability calculations, J. Geophys. Res., 109, B05304, doi:10.1029/2003JB002667.

Schoenball, M., C. Baujard, T. Kohl, and L. Dorbath (2012), The role of triggering by static stress transfer during geothermal reservoir stimula- tion, J. Geophys. Res., 117, B09307, doi:10.1029/2012JB009304.

---

## Author Comment (AC1) · 13 Jan 2017

**Earthquake static stress transfer in the 2013 Valencia Gulf (Spain) seismic sequence**

*L. Salo et al.*

**Authors' responses to RC1**

(AR) = Authors Response

(AC) = Authors changes being made in the manuscript

**General comments**

"In its current form the draft reads not very well and its major points are not clear and convincing enough. It needs a major revision to be worth publication. First of all, I think that this manuscript could significatively benefit from an overall restyling. I recommend a general and thourough improvement of the English (see Technical issues) and a more organized, concise and focused presentation of the method and results. The authors should be clerer in presenting and justifying their initial assumptions and stress out their goals (which is the final message and why is it significant in this context?). In this study the authors should bring into focus with more enphasis the specific analysis and result interpretation, which represents the innovative part of the study. Their results should be significative and presented in a convincing way. The methodology and the numerical code used for the static stress estimatimation are indeed already quite well known in the literature and do not represent novel tools."

(AR) First of all, we acknowledge that the chosen nomenclature for faults might not be the most appropriate. In the manuscript both "mapped" and "receiver" faults refer to the same faults (Main, East 2 and 4, and Montsia family), as opposed to the "source" faults (derived from Focal Mechanisms, FM, solutions). Here, we will only use the terms "previously mapped" (known geologic faults in the area, which do not slip in our modeling), and FM-derived or just FM (which do slip) faults to refer to them.

The manuscript's main goal is to assess the role of an earthquake triggering mechanism known as static stress transfer, during the seismic sequence of interest. Specifically, we try to answer the questions "was static stress transfer significant?" and "could it be responsible for the experienced events on its own?" (goal 1). In addition, and as a result of the computed stress changes, we evaluate 2) which of the previously mapped faults is/are more likely to have slipped based on the computed stress changes, and 3) if the experienced events would have shortened the occurrence of future earthquakes in the Main Fault.

To perform such an analysis it is necessary to know the orientation of the fault planes that hosted the earthquakes. FM's were computed for the 8 strongest events and thus are a usable source of information. Hence, our first assumption is that the 8 strongest events occurred on planes derived from the FM information (strike, dip, rake and location according to the solution). Computed FM depths for those events (beneath the reservoir) agree with the latest study focusing on earthquake location in the series (Gaite et al., 2016), but not with the previously mapped faults in the area, which are shallower. Thus, the previously mapped faults are used here as faults that receive stress only. The second assumption refers to the chosen slip plane, out of the 2 given by the FM solution. We chose all nodal planes to maximize ΔCS during the sequence (each time checking, for the selected slipping plane of event A, which of the two nodal planes of event B had, as a result, greater (positive) ΔCS and choosing that one). We model each of the source faults in COULOMB according to the previous reasoning. This procedure intrinsically presupposes that earthquake static stress transfer could indeed be responsible for the sequence, as we model the source planes based on its influence.(The remaining underlying assumptions of the analysis will be introduced accurately in the revised section of what is now #2).

However, owing to the fact that location has uncertainty, we cannot neglect the fact that (some of) the events could have responded to slip in the one or more of the here named previously mapped faults, should an error in locations and FM solutions exist; because of that, we provide assessment on which of the previously mapped faults would have been more likely to slip, based on obtained ΔCS.

As very well pointed out in this first comment, it is true that the method (analysis of Coulomb stress using Coulomb Failure Function) has long been established already and thus the important and new parts of the manuscript are the obtained results and specially its interpretation. Because of that, sections 4 (results) and 5 (discussion) were clearly separated in our first draft. The goal of section 4 is to describe the findings accurately so that they are well understood by the readers (because of that its subsections exist), and in section 5 we proceed to interpret the relevance of the results and argue our standpoint based on available references. Subsections were chosen to address goals 1-3 appropriately (the influence of static stress transfer, the hosting faults, and the shortening of the seismic cycle).

We developed subsections 2.1 and 2.2 to be synthetic (general method and formulae, very well known and documented already, e.g. King et al., 1994, for 2.1), whereas 2.3 and 2.4 contribute to increase the length of section 2 because they are particular of our study and we believed them necessary to justify our approach (2.4).

(AC) See next.

"The authors are studying the role of static stress redistribution due to earthquake mutual interactions during a seismic sequence that is probably induced by the injection of cushion gas for gas storage. In their analysis they are ignoring any influence that could come (and would probably be significant) from the triggering effect of pressure changes caused by the injection. This is a limiting but possible assumption if the focus of the study is on the role of earthquake interactions only (e.g. Baisch et al. 2009; Schoenball et al. 2012; Catalli et al. 2013). However, the authors have to stress out more clearly their specific goal and strong assumption from the beginning and touch the general issue of induced seismicity to give an overview on all the possible triggering mechanisms involved in such a case. The feeling after reading the text is that they comment their assumption of ignoring any other triggering effect during the final sections Discussions and Conclusions and it sounds like the study is incomplete due to shortage of data more than having a different initial aim."

In our analysis, the only way to introduce the effect of fluid pressure is by varying the effective friction coefficient (μ') in the Coulomb Failure Function (CFF), something that was done according to table C1. It can be understood from the Mohr-Coulomb failure criterion as the influence of a fluid within the fault (failure surface) with regard to the maximum shear stress that can be endured (slip threshold is lower with greater fluid pressure). However, as very well pointed out, this accounts for a static undrained condition, and not for fluid flow. We are indeed ignoring the influence of gas injections because 1) our goal is to assess the influence of a specific earthquake triggering mechanism other that pore pressure increase and 2) available information regarding gas injections is almost inexistent. According to your comment, we believe this is a limiting but possible approach.

We aimed to note our assumption (study of static stress redistribution on its own) in the introduction (#1) and provide a short comment on the induced seismicity topic in general in the shortcomings section (#5.4). We nevertheless did not develop it further in order to avoid introducing uncertainty and convolution to the study, which intends to focus on earthquake static stress transfer only. We agree that further discussion on the topic of the involved mechanisms should especially concern pore pressure generation.

Our goal is to transmit our findings regarding the specified triggering mechanism (case of study). Shortage of data is only noted as a fact.

(AC) The manuscript is being restyled accordingly, specifically #1 Introduction (regarding the objectives and assumptions made), #2 Method (reorganized to be clearer and further developed), # 4 Results and #5 Discussion (Focusing our text in the particular goals of the analysis, and introducing a general discussion on the topic of induced seismicity and triggers present in the series), and #6 Conclusions (presenting our most significant results in a concise and convincing way). Each sentence will be revised to assure its comprehension in English (according to Technical Issues). A Figure with a vertical profile of all modelled faults will be introduced so that the reader can rapidly distinguish the sources (FM) and previously mapped faults.

**Specific comments**

"I think that in this study there is an important confusing point: the calculation of ΔCFS made both on the *source fault planes* (section 4.1) and on the *mapped fault planes* (section 4.2). Is this distinction worthy and

why? I would avoid to say *source faults* when they are used thereafter also as receivers. Additionally, it is not clear which are the source planes used for estimating ΔCFS on the mapped faults; are also the mapped faults contributing to the cumulative ΔCFS as sources? The authors should be clearer in explaining which are the sources/receivers for each different case they present and list all relative information, as for example in Table B1. It is very difficult to interpret Figure 11 with current information; in particular, one does not easily know the time evolution of ruptures relative to the faults or fault's patches. Did for example the East 4 fault slipped at the beginning or in the middle of the sequence, or maybe at the end? Which are the sources contributing to the cumulative CFS estimated on top of the East 4 fault? How does ΔCFS on this specific fault evolve with time? This would help undertsanding its negative ΔCFS."

(AR)We believe the distinction between the FM-derived and the previously mapped faults to be of primary importance, and thus appreciate your comment on this issue. To understand the results and its interpretation, it is essential that we (the authors) manage to transmit the differences between them and our reasoning for doing it.

First, the source faults are based on FM information only. Their dimensions and geometry correspond to magnitude and FM solution of each of the 8 modeled events. They are deeper (mostly between 5 and 8 km beneath the seabed, according to the FM solution for each event) than the previously mapped faults, which only reach 3 km at most. The FM-derived faults have slip ≠ 0, because they correspond each one to one seismic event. Thus, they transmit a stress change to the neighboring regions, and during the sequence (before and after slipping) they are structures that can also receive stress changes from all FM faults. On the other hand, the previously mapped faults do NOT slip in our modelling. So, in our model, the Main, East 2 and 4, and Montsia family faults do not slip, and only receive stress due to slip in the FM-derived faults.

After the fault model in COULOMB with all FM-derived and previously mapped faults is built, the followed procedure consists in introducing the appropriate slip in FM 1 (to generate the first of the studied events), while all the others have slip 0. Afterwards, slip value in FM 2 is changed from 0 to its corresponding value, and thus we analyze the results after the second event (now both FM 1 and 2 with slip ≠ 0), and so on until all 8 FM have slipped. Slip in the previously mapped faults is always 0 and they do not contribute to ΔCS. We get the static stress changes after each of the 8 stages (the first corresponding just to the initial studied event, the last one cumulated after the 8 felt events), and thus can analyze its evolution and final state and generate figures 7 to 11, the latter being the cumulative result at the end.

Due to the fact that we aim to assess the importance of static stress transfer as an earthquake triggering mechanism, it is very important to study its evolution during the sequence on the FM faults (which are the sources in our modelling). At the same time, it is important to note the evolution of Coulomb stress values on the previously mapped faults so as to observe any relevant increase or decrease, but more important on these faults is their final stress state: we want *a)* to know if there is any evidence that could support slip on one of these faults during the sequence, and *b)* to quantify the shortening of the seismic cycle on the Main Fault. Regarding point *a)*, we are looking for generalized negative values along all fault patches. Because none of the previously mapped faults is allowed to slip in our modeling, a generalized negative value on one of them (as in East 4) results from a close FM-derived fault with similar geometry and location. Taking into account that source faults are placed at the exact location of the FM solution and that they have location uncertainty, our reasoning is that such a result provides evidence as to why one of the previously mapped faults (East 4) could indeed have been the one that slipped, thus being the cause of the last event (FM 8). Due to the different uses of analyzing stress variation on the FM-derived and on the previously mapped faults, we advocate for their distinction throughout the analysis (as in sect. 4.1 and 4.2).

(AC)The method section should be able to answer your questions (in the draft, sections 2.3.2 and 2.3.3 had this goal), and thus we agree we should rewrite it.

"Another crucial problem regards the interpretation of results in terms of ΔCFS: the authors conclude that the fault named East 4, which shows a negative cumulative ΔCFS at the end of the sequence (Figure 11), is therefore most likely to have slipped. However, another possible interpretation could be that the influence in terms of cumulative ΔCFS of all the previous events on the fault East 4 is negative, i.e. the fault is not

favored to slip by the cumulative stress redistribution. The stress drop on the fault might indeed have been caused also by its own slipping phase, but for a better understanding of this point one needs to know the fault's stress state before and after the time of its slipping in relation with the other events in the sequence (time evolution). In other words, ΔCFS on the East 4 fault is negative before or after its own rupture? This is also not clear from Figure 10. All these aspects should be clarified for a proper interpretation of results. Moreover, the fault is only partially visible in Figure 11."

(AR) We cannot formally exclude the fact that if a hidden structure exists at exact location where the FM fault was placed, then the East 4 fault could be farther from rupturing; but, given the proximity and similar characteristics, we believe our argument to be sound and the most likely given the geological investigations around the Castor UGS. Based on the modeled FM faults, the plane corresponding to the last event of the studied sequence is very similar, and closer in depth, to the East 4 fault (see figure 7, subplot 7, and table B.1). It is mainly because of the occurrence of this event that the resolved stress change onto the East 4 fault at the end of the sequence is negative in nature; thus, we argue that this event could indeed have taken place as a result of slip in the East 4 fault (each patch of the East 4 fault is large enough to produce a rupture of magnitude Mw 3.6).

(AC) Figure 10 will be modified so that the evolution on the East 4 fault is better appreciated, and Figure 11 will be plotted so that all patches on fault East 4 are easily seen. Section #5.2 in the discussion will be rewritten to clearly answer why we believe the East 4 fault could be the one that slipped.

"Referring to Figure 12, how are the best estimations of the parameters caculated/assumed? Why do the authors limit the analysis to just four parameters (strike, dip, rake and apparent friction)? Why do the authors perform this analysis referring to the mapped fault and not also the the so called source faults? The table that describe free parameter variations (now Table C1, which describes all parameters involved in the methodology) should be a dedicated table for the sensitivity analysis. Why do the authors vary the strike of the sources and not of the receivers? I think both have to be randomly perturbed within 20 degrees for understanding the sensitivity to the strike angle (the same for dip and rake). They might reproduce this way a large number of realizations for a more probabilistic based sensitivity analysis. The same concept would apply also for the nodal plane, depth and friction coefficient."

(AR)Regarding the previously mapped faults, we model the strike and dip (best estimation) from the references cited in 2.3.3, and needed rake for ΔCS modelling is inferred from FM solutions as we assume those events to result from faults slipping according to the regional stress. To estimate the rake, we calculate the mean value for each of the two nodal plane families (two main strike directions, see Fig. 13b) and assign the resulting mean rake to each of the previously mapped faults according to their strike (best estimation). Regarding μ', the recommended value (best estimation) for strike-slip or unknown faults is 0.4 (Stein et al., 1992; Toda et al., 2011).

We include strike, dip, rake and effective friction (plus depth for the Main Fault) in the sensitivity analysis, as they are ones subject to greater uncertainty in our study. We consider both the Young's modulus and Poison ratio to be well constrained, and the regional stress value does not influence the performed calculations (except for Fig. 13a), which focus on the stress changes between faults with previously defined characteristics (King et al., 1994). Background stress information is however included in table C.1 because it appears in the discussion (section 5 and Fig. 13a).

The sensitivity analysis was performed as well for the so called source faults (FM), but calculated variations were equal or smaller than the ones reported when the parameters on the previously mapped faults were varied. Moreover, due to the fact that the previously mapped faults have to be somehow simplified to include them in COULOMB (it was done according to explanations in 2.3.3) we believe the study of variations on their parameters to be more necessary than on the source faults.

Our goal with the sensitivity analysis is to address uncertainty by providing quantitative evidence on how results change when the main parameters are varied within likely ranges (magnitude of the variation), and on which parameter the analysis is most sensitive to. We varied the rake ± 20° instead of ± 10° because we believe its value to be less well constrained. We consider a variation of ± 20° for the strike and dip to be too

large (e.g. a fault that is reported to dip 50º would vary from 30º, which is a gentle dip, to 70º, not far from being sub-vertical). We agree that a complete probabilistic analysis would be interesting but we regard it to be out of scope in this study.

"Can the authors explain why the fact that the mean value in the East 4 fault is near -0.1 bar,supports the idea of this structure to have slipped (section 4.3.1 lines 22-24)?"

(AR) Because none of the previously mapped faults is allowed to slip in our modeling, a generalized negative value (most patches) on one of them (as in East 4) has been observed to result from a close source fault with similar geometry. Taking into account that source faults are placed at the exact location of the FM solution and that they have location uncertainty, our reasoning is that such a result provides evidence as to why the nearest previously mapped fault (here East 4) could have been the one that slipped.

"The final message about static stress transfer as triggering mechanism is not enough convincing (section 5.1): what is the take home message? The fact that the static stress acted only as a partial triggering factor is not a strong finding itself. The conclusions are dispersive."

(AR)Based on our findings, we believe that static stress transfer would have promoted the occurrence of the studied events. This is supported by positive $\Delta CS$ found on source fault planes, which are roughly of 0.1 bar in magnitude for the last 3 events (earlier studies have shown that values of this magnitude can promote seismicity). Thus, static stress transfer could be responsible, alone, for the occurrence of the last 3 events, but resolved values on the planes corresponding to the previous events are too small, according to cited references. Because of this, further comments are needed.

First, we investigated the orientation of the Optimally Oriented Fault Planes, and found that they are very similar in orientation to the ones obtained from FM solutions. This advocates for the studied events to have occurred according to background stress, and thus it seems logical that faults that moved were in fact about to move before the sequence and the injections (critically stressed). It supports the fact of a small perturbation already triggering the events. And secondly, because the pore pressure generation due to fluid flow did most likely influence the sequence (although our modeling cannot account for it), we think that static stress transfer acted as a triggering mechanism together with pore pressure increase. This reasoning is what the manuscript tries to transmit in section 5.1, and the reason why we refer to $\Delta CS$ only as a "partial trigger", although it reaches values near the assumed threshold of positive 0.1 bar for certain events.

(AC) As stated before, we will be more resolute and concise in our comments in the discussion (#5.1) and conclusions.

"The three first lines of Conclusions are confusing: for the East 4 fault a negative $\Delta CFS$ meant possible slip (see before), while here a positive $\Delta CFS$ resolved onto 7 of the 8 FM means possible destabilization. It looks like the authors are changing their point of view of the same process. They should be clearer and more consequential."

Bearing in mind that $\Delta CS = 0$ before the first studied earthquake, here that of September 24[th], we observe a positive $\Delta CS$ on 6 of the 7 remaining events (from second to last). For example, after the fifth earthquake, the plane that moves in the sixth event is positively loaded (see figure 7, subplot 5), and thus is destabilized by the previous five events. Then, after it slips, it has negative $\Delta CS$, which means it is far from slipping again (see subplot 6 in figure 7). Regarding the East 4 fault, it has negative values at the end of the sequence (far from slipping again) mainly due to the fact that the FM fault of the last studied event was close and with similar geometry; that is the reason why we believe it could have hosted at least one of the events, presumably the last one. We therefore think that our line of reasoning is the same throughout the whole analysis (positive stress changes = promote slip; negative stress changes = prevent slip).

**Technical issues**

Some examples of intricate, hard to understand sentences: p.2, lines 16-16; 22-23; 27-29 (here the authors seem to justify the fact that they use FM solutions as sources of Coulomb stress change. Why do they need

it? This is a very common assumption in the literature); p.3, lines 18-19 (explain which error is minimized and how). P.4, lines 4- 5: reduction of what? try to be clearer; lines 11- 13: explain better this concept of conservative or non-conservative assumption and which of the two you are following. P. 5, lines 3-4.

We highlighted the fact that FM solutions are used as sources of ΔCS because we work with the hypothesis that the faults that hosted (part of) the earthquakes could be unknown up to date. And to introduce that the known faults around the reservoir only receive ΔCS.

We cannot provide any other reply to the fact that our English and writing style are not clear enough, other than that we will work our best to improve them.

(AC)We will explain which errors is minimized and how in p.3 lines 18-19. Rewriting lines 4-5 and 11-13 in p.3 and 3-4 in p.5.

"Section 2 is probably too long and confused. I would suggest to respect the order resources/available data - methodology description – Coulomb model assumptions. I suggest to create a section on the methodology (Coulomb and seismic cycle) and one on the model assumptions (sources and receivers). The section about uncertainties should be independent and self-consistent. I think that the factors of uncertainty are several and regard both the methodologies involved in this study (the Coulomb stress estimation and the impact on the recurrence time). The authors should list them clearly and explain why do they analyse only some of them and how."

(AR) In this section we tried to limit its length as much as possible without compromising the understanding of the methodology and underlying assumptions. We thought it could be adequate to organize it starting with the equations (ΔCS and shortening of the seismic cycle), and follow with the modeling of faults and uncertainty in the assumptions and performed analysis. However, we agree with your suggestion to improve its comprehension and make the manuscript more up to the point.

(AC)We are reorganizing it as advised.

"Section 2.1: the authors allude to *various criteria to determine failure conditions* and then theydo not mention any of them. In general, if one refers to an issue, then one should at least spend a few words on that or omit it at all. Equation 1: why do the authors use the primes for *c* and σ? The simpler notation, the better. Later in equation 3 the prime disappears for Δσ and appears (correctly) for µ (but this causes confusion for the reader). CFF is defined but CS (line 7) is not. More generally on this section, it looks like that the authors are jumping from a formula to another for describing the concept of Coulomb failure criterion but they forget to give to the reader some simple elements for understanding a quite intuitive concept. If they want to start from the Mohr-Coulomb theory, they could then show a figure with a general description of the Mohr-Coulomb diagram. This would also be a good expedient for describing the possible effect (even though neglected in this study) of a pressure increase."

(AR)In equation 1, to introduce the Mohr-Coulomb failure criterion, we chose the typical notation used in soil and rock mechanics, selecting ' to indicate *effective* stresses instead of *total* stresses. We believe it to be appropriate given the fact that it is the effective stress that governs the behavior of geologic layers in the crust. We agree that it may be better to write it without primes, introducing the fluid pressure *u* as follows (regarding Eq. 1):

$$\tau = c + \mu(\sigma - u)$$

In the previous equation, the term $(\sigma - u)$ equals the σ' appearing in the manuscript version. It is true that our modeling can exclusively account for the effect of an undrained stress increase in the pressure, ultimately by introducing the effective friction coefficient (µ'). Thus, we should make it clear that fluid flow is not considered and so, pressure change by injections is out of scope.

 (AC)In section 2.1, we will provide examples of other criteria to determine failure conditions on rocks or omit them. In equation 1 we are introducing the simpler notation avoiding primes whenever possible. We are

defining ΔCS in this section as well (it was first introduced in page 2, line 25). We will develop the transition to equation 3 more accurately to avoid any confusion.

"The authors should be more careful in explaining the precise conditions under which equation 4 is valid, i.e. the pressure effect is taken into account only under the undrained condition (no fluid flow is considered) and the solely contribution to the pressure is given by the compressional terms of the stress tensor. They should remind that the pressure change due to the injection is still completely neglected in equations 3 and 4 and in general in their study."

(AC)We are proceeding as indicated.

"Why do the authors think that working with the software Coulomb 3.3 allows to deal with tridimensional complexity and not with other numerical models of the Coulomb stress redistribution? How do they justify their choice of calculating the stress change at 11 km depth (line 22, p.3)? From table B1 one can see that the 8 major events used as sources/receivers are all shallower than 5.9 km with an ecception of an event at 8.9 km. Additionally, Figure 5b does not help making a clear picture of the depths of the receiver faults. The depth of calculation of ΔCFS plays a significative role and should be discussed with greater attention."

(AR)We did not intend to say that other numerical models are not suitable, but that COULOMB is appropriate. We justify our choice on the fact that the deepest source is located at 11 km, responding to the FM solutions in table B.1. We should probably have indicated it better. The used depth in our modeling corresponds to the FM solution depth (column 7 in table B.1), and not to the depth of the input location. We prefer the depth of the FM solution given the fact that it adjusts all waveform instead of the depth of the earthquake location (used as input), which is based on phase picking and is not very well constrained because of seismic network distribution in the area.

(AC) We are ensuring that the chosen depth for sources is noted clearly in the text and table caption.

"Section 2.2: I would add some more explanations and comments referring to equations 5 and 6. Both equations imply the assumption of the concept of the characteristic earthquake. This is a strong assumption and it is worth some more discussion. Equation 5 implies that the average stress drop is released from a characterstic earthqake only, so that $\Delta\sigma/T_r$ corresponds with the regional stressing rate; all other possible phenomena of stress release are ignored. Equation 6 on the other hand implies the existence of a characteristic length for the source of a characteristic earthquake. I think that the authors should briefly comment on these hypotheses (why do they think these hypotheses are realistic in their context? What are the limitations?)"

(AR) Our main focus in this part is to quantify the shortening of the seismic cycle for the greatest earthquake a particular fault (the Main Fault) can host. Because of that, our goal is to quantify the relative importance of the computed ΔCS on that particular fault, regarding a particular earthquake (characteristic tremor). We ignore other possible phenomena of stress release (such as smaller earthquakes during the expected period of strain accumulation) based on our goal, which is putting the magnitude of the perturbation (ΔCS) in the context of the characteristic earthquake stress release (as previously done in Baisch et al., 2009 to cite an example). Indeed, In Equation 6 the assumption of fault geometry has its importance (greater length for the same rupture area results in lower stress drop).

(AC) We are working on a better justification of the use of equations 5 and 6, commenting the underlying assumptions and limitations.

"In general, I find that sometime in this draft the references to other studies for justifying or explaining some assumptions are ambiguous. For example, equation 5, p. 4: neither in *Harris* (2000) nor in *Baysch* et al. (2009) I can explicitly find any reference to this equation but solely hints to the idea that a stress change can affect the recurrence time of large earthquakes. I would find a reference as for example to the study of *Parsons* (2005) and references therein much more focused on the problem"

(AC) We are reviewing our given references and improving them whenever needed, specifically in this section 2.2 and generally throughout the whole text. We will try to avoid any ambiguity by using the appropriate citations.

"The *Main Fault* cited for the first time in correspondance of equation 9 and then in Figure 9 should be clearly defined. Please also define *SR*."

(AC) Proceeding as indicated.

"Section 2.3.1 can be merged with Data and Resources."

(AC) Proceeding as indicated.

"Section 2.3.1: rather than giving information on details about the input/output data of the method FMNEAR used for FM estimations, I would focus on the approach itself and the reliability of the method. ΔCFS estimations are indeed very sensitive to the FM solutions used for sources and receivers and their relative hypocentral distance. Why do the authors decided to use FM solutions obtained via FMNEAR? Why did they not use solutions already published in the cited literature (*Frontera et al.* (2013); *ING* (2013) and *Cesca et al.* (2014)? How sensitive are their results to the different solutions?"

(AR) We decided to select a method based on waveform modeling (rather than wave polarity) to improve reliability, and one that is stable for moderate earthquakes. This method seemed to fit well with our requirements and its quick online service was a plus, specially bearing in mind that finding a high-confidence solution for events with magnitude less than 4 is usually demanding.

We decided to generate new solutions based on the fact that the ICGC (Catalan Geologic Survey) had published in its bulletin of 2013 (ICGC, 2015) the earthquake locations for those events. Those locations had been found with a velocity model that worked well with registered waveforms, especially those from the closest sensors. We thought that this information was new, valuable and probably contained better constrained locations (earthquake location is used in the FM computation). Moreover, FM solutions are similar with the previous ones (see table B.1).

FM solutions are sensitive to input earthquake location and computed ΔCS depends on the FM solution. We therefore selected the best FM solution for each event and did not include more than one FM solution for a particular event in the analysis.

(AC) We are providing more insight in how sensitive are our results to different FM solutions in the revised version.

"Section 2.3.2: in the first paragraph of this section the English and wording needs a substantial improvement. The method for selecting the nodal plane is unclear. However, I do not agree/do not understand the idea that the nodal plane is selected referring to the highest ΔCFS. The authors need to justify their assumption. An alternative idea could be to quantify the difference when using a different nodal plane solution for each event or by using random extrapolated nodal plane solutions for a more probabilistic approach. P.5 from line 32: are the authors explaining how do they estimate the slip of each event? Does it come from the magnitude? It is unclear."

(AR) Based on the fact that we believe (hypothesis of the study) static stress transfer by neighboring earthquakes to be able to generate new events, we check ΔCS due to the previous event(s) on both nodal planes of event X, and select the one on which the ΔCS is greater (There is not any geologic information at the FM resolved depths, except for the last one, to discriminate between nodal planes; and, both nodal plane families are similarly oriented with regard to optimal orientations based on background stress). But, because of by doing that we needed to check each time both nodal planes, we could quantify the difference when using the other nodal plane (this is noted at the beginning of section 5.1, although only from a qualitative point of view).

Line 32 (P.5) and onwards: yes. We have the magnitude of each event, so we can know the seismic moment and therefore which is the value of slip needed to generate such a magnitude, given an area and shear modulus (working equation 10).

(AC) We will make sure that line 32 in p.5 and onwards is clear.

"Section 2.3.3: what I would need to understand clearly here is on which receiver planes do the authors estimate ΔCFS and why. From this paragraph one thinks that the FM solutions are used as sources and the geological faults as receivers. However, already in the abstract the authors say that :"…the evolution of static stress is quantified both on fault planes derived from focal mechanism solutions…and on the previously mapped structures…". The issue is touched then again in section 4. I think that the authors need to be more concise and precise on this assumption from the beginning and explain the reason of their assumptions."

(AR) This is, as indicated, an essential issue. ΔCS are estimated both on the FM-derived and on the previously mapped faults, along a sequence that is characterized by the slip of each nodal plane (each time for one of the 8 events). The previously mapped faults receive stress only, and FM-derived faults both transmit and receive stress. We estimate ΔCS on ALL FM and previously mapped fault planes. Each source fault has one patch, and the evolution of ΔCS on each source plane is studied along the sequence (Fig. 8). Each previously mapped fault is divided into various patches, and the ΔCS along the sequence is investigated on every patch (figures 9 and 10).

The idea understood from this paragraph is correct: FM solutions are used as sources. However, before and after an event's occurrence on a particular FM fault, we do not remove it in our model, thus when another event (hosted by a different FM fault) strucks, the previous FM fault receives stress as well. That is the reason why FM faults act as both sources and receivers, whereas the previously mapped faults are only receiving stress after each event (they never slip in our computations).

The process works as follows: first, we generate an input model with 8 FM-derived faults (selected nodal planes) and the 7 previously mapped faults. Following, slip is introduced along the first FM fault (first earthquake takes place), and ΔCS is computed onto each one of the 15 modeled faults. Then, slip is introduced along the second FM fault (second earthquake takes place), and ΔCS is computed onto all fault planes, and so on until the 8 main events have taken place. The model always contains 8 FM and 7 previously mapped faults. We need the FM faults to both generate and receive stress so as to test our hypothesis of static stress transfer as an earthquake triggering mechanism in this seismic sequence (goal 1), and the previously mapped faults to receive stress (goals 2 and 3).

Perhaps the problem comes from the chosen nomenclature, because the FM faults here both generate and receive stress, as opposed to most studies where the source fault merely generates slip and then ΔCS is studied on another fault, namely the receiver one.

(AC) We are making it clear enough on the method section to leave no room for doubts. We will add a table with sources (FM) in one entry (e.g. rows), and receivers (both FM and previously mapped faults) in another one (e.g. columns).

"Section 2.4: this section is not complete. A reader needs to understand which are the parameters at play; which of these parameters influence uncertainties the most; how do the authors perform a comprehensive sensitivity analysis and major findings. To be more concise, tha authors may merge section 2.4 directly into section 4.3."

(AC) We are revising it as suggested.

"Section 4.4: first line: the poor reader needs to check the two tables ahead and equation 5 behind to understand what you are discussing about. The tables can be merged together. The case *Best* should be better defined and justified. The tables are not easily comprehensible. Where are *the results for the best estimate* plotted? There is no reference to a Figure."

(AR) The results for the best estimate (according to most likely strike, dip, rake and μ') are shown in bold in the tables. They are not plotted in any figure as we thought the shortening is explained enough in % of shortening and in years shortened out of the earthquake's recurrence time.

(AC) We are working on a figure to represent the shortening and complement the table (tables 1 and 2 will be converted into one as suggested). We will probably use a bar graph or a scheme.

"Section 5.2, the first lines are incomprehensible."

(AC) Our apologies for that. We are reformulating them.

"In Figure 1 would be interesting to see also the injection rate."

(AR)Yes, certainly. Nevertheless, the only reliable information we have from scientific works comprises both the duration of the third injection phase and an estimation of the total introduced volume of fluid during this stage. Further discussion would undoubtedly be fostered if the injection rates and introduced volume per day were available.

"Figures 5 and 11 can be merged together into an unique Figure."

(AC) Proceeding as indicated.

"Figure 7 is not clear and one cannot follow the relative comments reported in section 4.1. Projecting ΔCFS on the FMs can just give a rough idea of the stress change (positive/negative) and information on the spatial distribution is lost. Why do the authors decided for this kind of presentation? Which is the message they want to give through this figure?"

(AR) We plotted this figure with the goal of clearly displaying which nodal plane was selected (pink line on the FM), the relative location (on the XY plane) and time of occurrence (FM number) of each earthquake considered. Moreover, the goal is to provide a first approach to the nature of the resolved ΔCS onto the plane of each event before it occurred, highlighting the nature of the variation (positive or negative).We believe it compliments itself with the following figure, which is better for quantitative purposes.

(AC)The previously mentioned table will clear doubts regarding this figure.

"Figure 8: which are the events causing a ΔCFS on the FM of the event occurred the 09/24?"

(AR) This is the first felt event in the series and the first considered one in our study. Because of that, before its occurrence the ΔCS on either nodal plane is 0. We selected the one that caused greater ΔCS on one of the two possible nodal planes for the next event.

Before this first considered event, quite a remarkable number of earthquakes had already taken place in the area (see figure 2), but no FM solution is available for any of them (magnitudes are too small). Thus, this first event cannot be justified by our modeling. After its occurrence, the 7 remaining events will introduce some Coulomb stress variation onto its plane, but it will be minor compared to the stress drop after it slipped on September 24[th] (negative ΔCS).

"Figure 11: report in the figure, close to the corresponding patch, the identifying letter+number already reported in Figure 10."

(AC) proceeding as indicated.

"Tables B1 and C1 are fundamental for the interpretation of the methodology and results, why are there in the appendix?"

(AR)Although the tables B1 and C1 are in fact necessary,  and we agree they could be part of the main text, we think they are better placed in the appendix because chosen FM solutions and ranges of parameters are already commented in the text. The tables provide as well lots of supporting information (previously

published FM solutions or references in C1) which can make the manuscript less swift to read if placed in the main text.

(AC) We will consider the option of placing them in the main text (in the revised version), bearing in mind the length of the manuscript.

**References**

Baisch, S., D. Carbon, U. Dannwolf, B. Delacou, M. Devaux, F. Dunand, R. Jung, M. Koller, C. Martin, and M.Sartori.: Deep heat mining basel: seismic risk analysis, SERIANEX Group, Departementfür Wirtschaft, Soziales und Umwelt des Kantons Basel-Stadt, Basel, Switzerland, 2009.

Gaite, B., Ugalde, A., Villaseñor, A., and Blanch, E.: Improving the location of induced earthquakes associated with an underground gas storage in the Gulf of Valencia (Spain), Physics of the Earth and Planetary Interiors, 254, 46-59, 2016.

King, G. C. P., Stein, R. S., and J. Lin: Static Stress Changes and the Triggering of Earthquakes, Bull. Seism. Soc. Amer., 84(3), 935-953, 1994.

Stein, R. S., King, G. C., and Lin, J.: Change in failure stress on the southern San Andreas fault system caused by the 1992 magnitude= 7.4 Landers earthquake, Science, 258(5086), 1328-1332, 1992.

Toda, S., Stein, R. S., Sevilgen, V., and Lin, J.: Coulomb 3.3 graphic-rich deformation and stress-change software for earthquake, tectonic, and volcano research and teaching-user guide (No. 2011-1060).US Geological Survey, 2011b.

ICGC. Butlletí Sismològic 2013, Departament de Territori i Sostenibilitat, Generalitat de Catalunya. Barcelona (Catalunya), 2015.

---

## Referee Comment (RC2) · Anonymous Referee #2 · 17 Feb 2017

I think that the problem raised in the manuscript is potentially of broad interest for SE readers and the scientific community, and should be considered for publication. However, reading this manuscript I am confused about that if it is at all possible in the form presented here. First of all the aim of the work is not clearly presented and the final conclusions are also not stressed and strong enough. Authors admit that future studies are needed when additional data will be available. But my main concerns is the proposed methodology to check the contribution of the mapped faults in the analyzed seismic sequence in the Valencia Gulf. I understand the first component of work to consider possible cause of interactions among seismic events as static stress transfer. Authors focus on the cumulative changes in stress due to the consecutive seismic events in the analyzed series. The cumulative stress changes are calculated after the occurrence of each event according to location and faulting type of the next event

in the series. Although for this part I have some comments which I provide below, I think this part after improvements would be ready for publishing. The problem is with the two other goals, if I identified them properly: the contribution of mapped faults in the seismic sequence and the contribution of static stress changes in the seismic cycle of these faults. If the Authors assumed to consider their own Focal Mechanisms (FM) and depths of events (from 3 to 11 km) how is the sense to resolve the stresses of these events from the depth of these events on the mapped faults planes at the depth of these faults and at the same time hypothetically expecting that maybe these mapped faults contribute in the slip of the whole sequence. In my opinion if they could contribute they first should correlate with the parameters of the following seismic event in the sequence and second, if the range of the depths of events in the sequence is consistent and similar FM of events are as we see in Table B1, mapped faults had to experience the Coulomb Failure Function changes from the events at similar depths as they are. We know that the depth factor plays very important role in the CFF changes (DCFF) analyses. The consistency of the depth of events in the sequence is easy to be proven by the normalized signal cross-correlation (e.g. Schaff P. and Waldhauser F., 2005). Looking at the Table B1 one can notice that the FM of events are not so different to each other. Based on the idea that signals of events with close hypocenters and similar FM recorded on the same station are very similar, the signals cross- correlation analysis may indicate the possible differences in recorded signals either due to events' different depths or focal mechanisms. Moreover, this analysis may reveal some highly correlated pairs within events group. Did the Authors perform such kind of analysis? The same problem I see with the cumulative CFF changes impact on the seismic cycle of the mapped faults. If the mapped faults experienced CFF changes due to events on shallower depths the values of CFF changes would be quite different. More detailed comments in attached supplement.

Please also note the supplement to this comment:
http://www.solid-earth-discuss.net/se-2016-146/se-2016-146-RC2-supplement.pdf

[Figure]

**Supplement:**

Review of the paper "Earthquake static stress transfer in the 2013 Valencia Gulf (Spain) seismic sequence" by L. Salo et al. submitted to Solid Earth

I think that the problem raised in the manuscript is potentially of broad interest for SE readers and the scientific community, and should be considered for publication. However, reading this manuscript I am confused about that if it is at all possible in the form presented here. First of all the aim of the work is not clearly presented and the final conclusions are also not stressed and strong enough. Authors admit that future studies are needed when additional data will be available. But my main concerns is the proposed methodology to check the contribution of the mapped faults in the analyzed seismic sequence in the Valencia Gulf. I understand the first component of work to consider possible cause of interactions among seismic events as static stress transfer. Authors focus on the cumulative changes in stress due to the consecutive seismic events in the analyzed series. The cumulative stress changes are calculated after the occurrence of each event according to location and faulting type of the next event in the series. Although for this part I have some comments which I provide below, I think this part after improvements would be ready for publishing. The problem is with the two other goals, if I identified them properly: the contribution of mapped faults in the seismic sequence and the contribution of static stress changes in the seismic cycle of these faults. If the Authors assumed to consider their own Focal Mechanisms (FM) and depths of events (from 3 to 11 km) how is the sense to resolve the stresses of these events from the depth of these events on the mapped faults planes at the depth of these faults and at the same time hypothetically expecting that maybe these mapped faults contribute in the slip of the whole sequence. In my opinion if they could contribute they first should correlate with the parameters of the following seismic event in the sequence and second, if the range of the depths of events in the sequence is consistent and similar FM of events are as we see in Table B1, mapped faults had to experience the Coulomb Failure Function changes from the events at similar depths as they are. We know that the depth factor plays very important role in the CFF changes (DCFF) analyses. The consistency of the depth of events in the sequence is easy to be proven by the normalized signal cross-correlation (e.g. Schaff P. and Waldhauser F., 2005). Looking at the Table B1 one can notice that the FM of events are not so different to each other. Based on the idea that signals of events with close hypocenters and similar FM recorded on the same station are very similar, the signals cross-correlation analysis may indicate the possible differences in recorded signals either due to events' different depths or focal mechanisms. Moreover, this analysis may reveal some highly correlated pairs within events group. Did the Authors perform such kind of analysis? The same problem I see with the cumulative CFF changes impact on the seismic cycle of the mapped faults. If the mapped faults experienced CFF changes due to events on shallower depths the values of CFF changes would be quite different. Now more detailed comments.

1. Part of the results presented in the manuscript is based on the assumption of characteristic earthquake phenomenon. Could the Authors provide the justification of such approach?

2. Paragraph 2.3.1. The provided description of the web-service is to detailed and unrequired in the comparison to the other sections of manuscript.

3. Paragraph 2.3.2. I do not see the justification of the implemented approach to select the slipped nodal plane. As Authors stressed several times in the paper, the stress changes that are the cause of seismic events are also due to other factors than only DCFF. Seismicity accompanying technological activity results from changes in the stress field in the rock mass mainly due to this activity. If the rock mass are in highly pre-stressed conditions, even small stress perturbations can cause seismic events. Thus, it is not excluded that the plane in such sequence cannot experience the negative DCFF. Here the pore pressure changes modelling is not taken into account. The best would be to investigate all the possible plane scenarios and then to provide statistical based conclusions.

4. Distinguish between the two Paragraphs 2.3.2 and 2.3.3 is misleading since receiver faults are also source faults. Moreover, the nomenclature used to determine faults is also misleading because we have source, receiver, mapped and finally hosting faults. Could the Authors think over this issue how to simplify the information for easier understanding the contents.

5. Paragraph 2.4. Authors consider the particular impact of the input parameter uncertainties on the results. But the most relevant is the combined approach which may be achieved using synthetic samples and then statistical inference. Even if we focus on the sole effect due to particular parameter uncertainties there is no information how the calculations were performed. How many synthetic values was considered, or maybe only the worse and the best scenario. Fig. 12 presenting the results of this step of analysis could have a scale of DCFF more adopted to the range of the experienced distribution of DCFF.

6. Information from the Paragraph 3 should be incorporated into other Paragraphs.

7. Paragraph 4.1. Line 20. Indeed an empirical threshold for triggered natural seismicity of 0.01 MPa is usually used (e.g. Reasenberg & Simpson 1992; King et al. 1994). While many studies suggest that triggering requires a minimum stress change, the variation in this threshold spans an order of magnitude or more. In mining induced seismicity the minimum Coulomb stress change that influences the occurrence of future seismicity was 0.005 MPa; this triggering threshold was confirmed to be statistically significant (Orlecka-Sikora, 2009). Another example from mining induced seismicity in Deep Gold Mine in South Africa suggests also that seismic activity was triggered by mainshock in the areas where static stress increased not more than 0.01 MPa (Kozlowska et al., 2015). Ziv and Rubin (2000) and Ogata (2005), however, point out that triggering is not a threshold process. Hardebeck et al. (1998) suggest that any small stress change is capable of triggering and the existence of an apparent minimum triggering stress is connected with the sufficient number of triggered events to be detectable with dataset used.

8. Paragraph 5. Line 10-13. The papers e.g. Orlecka-Sikora, 2009; Kozlowska et al., 2016, open ICHESE Report describing the May-June 2012 Emilia sequence case provide the results of analysis of impact of the cumulative DCFF on the following events in the considered sequences.

9. What about the distribution of smaller seismic events? How they are distributed according to the DCFF due to stronger events and particular FM and mapped faults. The smaller events distribution may provide additional insights into the location of stronger events and their slipped plans.

10. The quality of the Figures 8-10 is not satisfied. The DCFF scale is missed on Fig. 10.

11. Paragraph 2.1. Line 23: an homogenous -> a homogenous; Paragraph 2.3.1. Line 2: solution consists on inverting -> solution consists in inverting; Line 3: whose information is -> that are.

References:

Hardebeck, J.L., Nazareth, J.J.&Hauksson, E. 1998. The static stress change triggering model: constraints from two southern California aftershock sequences, *J. geophys. Res.,* **103,** B10, 24 427–24 437.

International Commission On Hydrocarbon Exploration And Seismicity In The Emilia Region (ICHESE), Report on the hydrocarbon exploration and seismicity in Emilia Region, (2014, http://unmig.sviluppoeconomico.gov.it/unmig/agenda/dettaglionotizia.asp?id=175).

King, G.C.P., Stein, R.S. & Lin, J., 1994. Static stress changes and the triggering of earthquakes, Bull. seism. Soc. Am., 84, 935–953.

Kozłowska, M., Orlecka-Sikora, B., Kwiatek, G., Boettcher, M. S., Dresen, G. (2015) Nano- and picoseismicity rate changes from static stress triggering caused by a MW2.2 earthquake in Mponeng gold mine, South Africa, *J. Geophys. Res. Solid Earth*, 120, doi:10.1002/2014JB011410.

Kozłowska, M., Orlecka-Sikora, B., Rudziński, Ł., Cielesta, S., Mutke, G. (2016) Atypical evolution of seismicity patterns resulting from the coupled natural, human-induced and coseismic stresses in a longwall coal mining environment, *International Journal of Rock Mechanics and Mining Sciences* 86, 5-15, doi: 10.1016/j.ijrmms.2016.03.024.

Ogata, Y. (2005), Detection of anomalous seismicity as a stress change sensor, J. Geophys. Res., 110, B05S06, doi:10.1029/2004JB003245.

Orlecka-Sikora, B. (2010) The role of static stress transfer in mining induced seismic events occurrence, a case study of the Rudna mine in the Legnica-Glogow Copper District in Poland, Geophys. J. Int., 182, 1087–1095

Reasenberg, P.A.&Simpson, R.W., 1992. Response of regional seismicity to the static stress change produced by the Loma Prieta earthquake, Science, 255, 1687–1690.

Schaff P. and Waldhauser F. (2005). Waveform Cross – Correlation – Based Differential Travel – Time Measurements at the Northern California Seismic Network, Bull. Seismol. Soc. Am. 95, no.6, pp 2446-2461, doi: 10.1785/0120040221.

Steacy, S., J. Gomberg, and M. Cocco (2005), Introduction to special section: Stress transfer, earthquake triggering, and time-dependent seismic hazard, J. Geophys. Res., 110, B05S01, doi:10.1029/2005JB003692

Ziv, A., and A. M. Rubin (2000), Static stress transfer and earthquake triggering; no lower threshold in sight?, J. Geophys. Res., 105, 13,631– 13,642.

---

## Author Comment (AC2) · 24 Feb 2017

**Earthquake static stress transfer in the 2013 Valencia Gulf (Spain) seismic sequence**

L. Salo et al.

AR = Authors' Response
AC = Authors' changes being made in the manuscript
* * *
**1   General comment**

"I think that the problem raised in the manuscript is potentially of broad interest for SE readers and the scientific community, and should be considered for publication. However, reading this manuscript I am confused about that if it is at all possible in the form presented here. First of all the aim of the work is not clearly presented and the final conclusions are also not stressed and strong enough. Authors admit that future studies are needed when additional data will be available. But my main concerns is the proposed methodology to check the contribution of the mapped faults in the analyzed seismic sequence in the Valencia Gulf. I understand the first component of work to consider possible cause of interactions among seismic events as static stress transfer. Authors focus on the cumulative changes in stress due to the consecutive seismic events in the analyzed series. The cumulative stress changes are calculated after the occurrence of each event according to location and faulting type of the next event in the series. Although for this part I have some comments which I provide below, I think this part after improvements would be ready for publishing."

(AR) After both reviews, we as authors acknowledge the need to restyle the whole manuscript in order to present our method, assumptions and results more clearly. In addition, we will do our best to be concise and stress out our conclusions. The identification of the first goal of the study is correct (we aim at quantifying $\Delta$CS after the occurrence of each event; in exception of the first event, cumulative stress changes are presented).

(AC) We are restyling the manuscript to both present a clearer approach to the methodology used and assumptions made, and also results and discussion. In the introduction, the goals of the analysis will be clearly indicated. We believe in our analysis to provide new information of interest regarding the 2013 Valencia Gulf seismic sequence, and thus are confident of its value; after an overall restructuring, clearer writing and improved approach to specific sections, we expect it to have the quality needed to be submitted for a final evaluation in this journal.

"The problem is with the two other goals, if I identified them properly: the contribution of mapped faults in the seismic sequence and the contribution of static stress changes in the seismic cycle of these faults. If the Authors assumed to consider their own Focal Mechanisms (FM) and depths of events (from 3 to 11 km) how is the sense to resolve the stresses of these events from the depth of these events on the mapped faults planes at the depth of these faults and at the same time hypothetically expecting that maybe these mapped faults contribute in the slip of the whole sequence. In my opinion if they could contribute they first should correlate with the parameters of the following seismic event in the sequence and second, if the range of the depths of events in the sequence is consistent and similar FM of events are as we see in Table B1, mapped faults had to experience the Coulomb Failure Function changes from the events at similar depths as they are. We know that the depth factor plays very important role in the CFF changes (DCFF) analyses. The consistency of the depth of events in the sequence is easy to be proven by the normalized signal cross-correlation (e.g. Schaff P. and Waldhauser F., 2005). Looking at the Table B1 one can notice that the FM of events are not so different to each other.

Based on the idea that signals of events with close hypocenters and similar FM recorded on the same station are very similar, the signals cross- correlation analysis may indicate the possible differences in recorded signals either due to events' different depths or focal mechanisms. Moreover, this analysis may reveal some highly correlated pairs within events group. Did the Authors perform such kind of analysis?"

(AR) The two other goals (second and third) are correct. There is one clarification which is worth noting, although it may be unnecessary here; previously mapped faults DO NOT slip in our modeling. Due to the fact that most of the seismicity seems to have taken place in between 5 and 8 km of depth, with both latest locations [1], and this study's best FM solutions for each event being consistent in this sense, we use FM results to place the sources. However, taking into account that one event is located at 3 km of depth and 2 others are quite close (resolved at 5 km), and also that owing to injections shallower locations should be expected, we investigate the cumulative stress changes along the whole sequence in the previously mapped faults as well to assess whether there is evidence that could support slip on these previously mapped structures. This does not mean we believe they did actually slip (we agree with conclusions by [1]), but rather, that this possibility should not be completely discarded yet. From our standpoint, if some FM solutions are close enough in depth and in geometry to cause relevant negative $\Delta$CS onto one (or more) of the previously mapped structures, the simplest assumption is to think that the event could have occurred on the mapped fault. We take this position based on location uncertainty due to available seismic network distribution, and different depths at which FM solutions have been resolved (compare depths reported by [2] and the later analysis by [1]). We believe this assumption to be compatible with the usage of FM solutions to place the sources for our modeling, which we believe to be deeper than the previously mapped faults.

Regarding the Signals Cross Correlation (SCC) analysis, it was not performed here. FM solutions were chosen for each event after varying input depth and selected waveforms, as described in the draft, and the highest-confidence solution was chosen for each event. Hence, we believe the presented FM solutions to be reliable enough as they are. Moreover, SCC was performed by [1], who found solution depths very similar to this study (For example, they place two of the M > 4 events at 6 km of depth and another one at about 10, which is almost the same we obtained in the presented FM solutions). SCC in their analysis allowed them to reduce scatter in hypocentral locations; their improved locations show almost all seismicity in between 3 and 10 km of depth, with most of it occurring at depth = 6 km. Thus, depths of the located events and previously mapped structures could not be linked.

(AC) Even if we do not rule out the possibility that some of the previously mapped faults could have slipped based on resolved $\Delta$CS, the fact that the previously mapped faults experienced the Coulomb Failure Function changes from the events at similar depths (as they are placed from FM solutions) is also possible. We agree that this should be better clarified in the discussion. Moreover, it will be stressed that even though the remote possibility of slip on one of the previously mapped structures exists, our study globally supports deeper fault sources. A cross-section in which *FM* and *previously mapped* faults all appear will be included to highlight depth differences.

"The same problem I see with the cumulative CFF changes impact on the seismic cycle of the mapped faults. If the mapped faults experienced CFF changes due to events on shallower depths the values of CFF changes would be quite different."

(AR) We agree that the values of $\Delta$CS would be quite different, should the mapped faults have experienced stress transfer from shallower events. But, as commented before, we consider our FM solutions to be reliable at the placed depth. We expect the 8 main events to be representative of, at least, the strongest phase of the sequence (which started after injections had been halted). These events occurred a week or two later than the injections halt and this second phase has been acknowledged to be different in nature than the first one (e.g. [2]). No FM solutions are available for events of smaller magnitudes at the moment. Thus, it should be reasonable to expect that stress perturbations due to smaller events at similar depths would not change computed values onto the previously mapped faults enough for the conclusions drawn from the performed analysis to be significantly different. Believing in this hypothesis, we performed the calculations to obtain the shortening of the seismic cycle onto the Main Fault plane.

(AC) We are ensuring the hypotheses and limitations of the performed approach are clearly

written.

**2 Detailed comments**

"1. Part of the results presented in the manuscript is based on the assumption of characteristic earthquake phenomenon. Could the authors provide the justification of such approach?"
(AR) In our analysis, we investigate the stress state as resolved onto a particular fault plane, and the shortening in its seismic cycle as a result of computed $\Delta$CS. The characteristic earthquake theory implies that earthquake occurrences on single faults and fault segments do not follow the Gutenberg-Richter relationship. This hypothesis was initially bolstered by paleoseismicity data; later, other works based on seismological and geological observations reinforced it (discussion is provided in [3, 4]). This hypothesis has been previously used in studies addressing the effect of $\Delta$CS on earthquake recurrence times, both due to large earthquakes and also in anthropogenic seismicity assessment. In addition, previous works in the Valencia Gulf used this approach as well ([5, 6]; we apologize for these references being in Catalan and Spanish respectively).
We believe it to be the most logical hypothesis to be done in this case so as to be consistent with previous works conducted in this area and in this context.
(AC) We are ensuring that our approach is properly justified in the section dedicated to this part's methodology, which will be longer than this reply and supported by references.

"2. Paragraph 2.3.1. The provided description of the web-service is to detailed and unrequired in the comparison to the other sections of manuscript."
(AC) We will take it into account and modify it.

"3. Paragraph 2.3.2. I do not see the justification of the implemented approach to select the slipped nodal plane. As Authors stressed several times in the paper, the stress changes that are the cause of seismic events are also due to other factors than only DCFF. Seismicity accompanying technological activity results from changes in the stress field in the rock mass mainly due to this activity. If the rock mass are in highly pre-stressed conditions, even small stress perturbations can cause seismic events. Thus, it is not excluded that the plane in such sequence cannot experience the negative DCFF. Here the pore pressure changes modelling is not taken into account. The best would be to investigate all the possible plane scenarios and then to provide statistical based conclusions."
(AR) This is a critical issue and we are grateful for your comment. This is perhaps one of the most challenging parts in this study, and we have taken into consideration the following:

  a. The first option that is always explored is comparing both conjugate planes with previously mapped structures to see if one can be discarded. This is hardly feasible here due to 1) the fact that similar structures compatible with strikes of both Nodal Planes (NP) are present and 2) depth difference for most of the solutions.

  b. Another option that has been used in studies of anthropogenic seismicity [7], is using the hypocenter distribution of adjacent events to choose the causative nodal plane "by eye". Here, seismicity cloud density around each FM solution and location uncertainty make it inadvisable to use this approach.

  c. A possible alternative is using background stress to determine Optimally Oriented Fault Planes (OOFP) and then chose the NP, for each pair, that is closer to OOFPs. It was first discarded after a rough look of stress orientations based on which both conjugate planes for each FM were similarly oriented. However, we are currently quantifying it and this will be used in our revised selection of nodal planes.

  d. Under the assumption of events triggered by fluid overpressures, the Critical Pore Pressure (CPP) criterion may be used to select the causative nodal plane (e.g [8]). We, as authors, are not favorable to using this alternative on its own, due to the fact that almost no information regarding reached overpressures has been made public (using this method for

NP selection implies assuming that the events took place as a result of fluid overpressures). In spite of that, an evaluation of the selected NP using this criterion is likely to be included in our revised version.

**e.** Our hypothesis in the discussion manuscript involved considering earthquake static stress as a destabilizing trigger; thus, the selection of each NP was made after a test-and-error analysis in which both alternatives for each FM of a particular event at time *x* were explored, and the one with higher $\Delta$CS as a result of all events with FM solution up to time *x-1* was selected.

(AC) We agree with your comment regarding the consideration of other present factors that may have played a role in the activation of fault planes. Because of that, alternative **e.** should be contrasted with other options so as to provide a broader range of plane scenarios. We are going to include information from at least **c.** in our revised version.

"4. Distinguish between the two Paragraphs 2.3.2 and 2.3.3 is misleading since receiver faults are also source faults. Moreover, the nomenclature used to determine faults is also misleading because we have source, receiver, mapped and finally hosting faults. Could the Authors think over this issue how to simplify the information for easier understanding the contents."
(AR) We chose "source" and "receiver" terms based just on whether the fault slipped or not in our modeling. Thus, selected NP from FM solutions were named "source" and the previously mapped structures in the area "receiver".
(AC) We agree that, even with the aforementioned terms being chosen with our best intentions, nomenclature should be rethought. This is specially due to the fact that both FM faults and previously mapped structures receive stress. We are going to name *FM-derived* or just *FM* faults the fault planes selected from FM information, and *previously mapped* or just *mapped* faults will be used instead of receiver. It will be noted that both receive stress, and that only *FM* faults slip in the performed modeling. We are considering to include a table to better note this distinction.

" 5. Paragraph 2.4. Authors consider the particular impact of the input parameter uncertainties on the results. But the most relevant is the combined approach which may be achieved using synthetic samples and then statistical inference. Even if we focus on the sole effect due to particular parameter uncertainties there is no information how the calculations were performed. How many synthetic values was considered, or maybe only the worse and the best scenario. Fig. 12 presenting the results of this step of analysis could have a scale of DCFF more adopted to the range of the experienced distribution of DCFF."
(AR) The final stress state on the previously mapped faults was first computed by using a *best estimation* of the parameters. This corresponds to geometry (strike, dip and rake) and depth as given by the selected NP out of each FM, and $\mu$ (effective friction coefficient) of 0.4. Afterwards, variations in the geometrical parameters and depth of each NP were carried out to complete the sensitivity analysis. The strike was varied up to $\pm15°$ (this should be corrected in table C1, our apologies) in 5° increments (e.g. NP strike, NP strike + 5°, NP strike + 10°, NP strike + 15°, NP strike -5° and so on), the dip up to $\pm10°$ in 10° increments, the rake up to $\pm20°$ in 10° increments, and final result with $\mu$ 0.2 and 0.6 was also computed.
Depth of all FM solutions was also switched 1, 2 and 3 km upwards, until the shallowest solution reached "surface depth". This part focused on the observed variations in the Main Fault. In spite of the fact that computed values onto particular patches augmented (distances were minor), the total area (total number of patches) of the Main Fault subject to positive $\Delta$CS did not remarkably change.
Our analysis is focused on the "best estimate" of the parameters. The sensitivity analysis was performed as a complementary tool to investigate sole effects due to particular parameter uncertainties, as indicated in your comment. Each variation was introduced on its own and the final stress state was computed just for each variation (e.g. 7 different calculations were performed regarding the strike). After each calculation step, the maximum, minimum and average (computed from all patches of a particular mapped fault) values of $\Delta$CS were obtained, and this is how Fig. 12 was plotted. In this case, the selected variations were chosen "by logic" and

not owing to any specific calculation or reference. We aimed at maintaining the nature of each solution. As indicated in our response to RC1, we varied the rake $\pm20°$ instead of $\pm10°$ because we believe its value to be less well constrained. We consider a variation of $\pm20°$ for the strike and dip to be too large (e.g. a fault that is reported to dip 50° would vary from 30°, which is a gentle dip, to 70°, not far from being sub-vertical).

(AC) We will revise our sensitivity analysis and probably include the same range of variation for all geometric parameters (in this case $\pm15°$ will be used for strike, dip and rake). We agree that a further study of FM depth and its specific effect onto each mapped fault would be interesting but believe it to be out of scope in this study (which considers a best estimation of the parameters), as would be a complete probabilistic analysis.

We are not sure about changing the scale of $\Delta$CS in Fig. 12 due to the fact that it was already adopted to ease the comparison of the experienced distribution of values, but we will consider it.

"6. Information from the Paragraph 3 should be incorporated into other Paragraphs."
(AC) Proceeding as indicated.

"7. Paragraph 4.1. Line 20. Indeed an empirical threshold for triggered natural seismicity of 0.01 MPa is usually used (e.g. Reasenberg & Simpson 1992; King et al. 1994). While many studies suggest that triggering requires a minimum stress change, the variation in this threshold spans an order of magnitude or more. In mining induced seismicity the minimum Coulomb stress change that influences the occurrence of future seismicity was 0.005 MPa; this triggering threshold was confirmed to be statistically significant (Orlecka-Sikora, 2009). Another example from mining induced seismicity in Deep Gold Mine in South Africa suggests also that seismic activity was triggered by mainshock in the areas where static stress increased not more than 0.01 MPa (Kozlowska et al., 2015). Ziv and Rubin (2000) and Ogata (2005), however, point out that triggering is not a threshold process. Hardebeck et al. (1998) suggest that any small stress change is capable of triggering and the existence of an apparent minimum triggering stress is connected with the sufficient number of triggered events to be detectable with dataset used."
(AR) We certainly appreciate this comment, as well as the following one. We will revise the references provided to foster our discussion regarding this part.

"8. Paragraph 5. Line 10-13. The papers e.g. Orlecka-Sikora, 2009; Kozlowska et al., 2016, open ICHESE Report describing the May-June 2012 Emilia sequence case provide the results of analysis of impact of the cumulative DCFF on the following events in the considered sequences."
(AR) We will study the results obtained from the analysis of the impact of cumulative $\Delta$CS in the events that follow in a sequence.
(AC) Comments regarding the references provided will be made in the discussion (section 5 in the manuscript).

"9. What about the distribution of smaller seismic events? How they are distributed according to the DCFF due to stronger events and particular FM and mapped faults. The smaller events distribution may provide additional insights into the location of stronger events and their slipped plans."
(AR) As indicated in the response to comment 3, it is true that the smaller events distribution may provide useful information regarding the selection of the NP. However, the problem in this case is that location uncertainty for smaller events, which are logically much more numerous, is too high to use for this purpose. The general cloud of events seems to be distributed NW-SE, but no particular alignment of seismicity that could be used for this purpose (high density of events in a constrained area around the FM placements) has been reported either (see [1]).
(AC) We are accounting for the distribution of smaller events when it comes to the general interpretation of the results, but not for NP selection.

"10. The quality of the Figures 8-10 is not satisfied. The DCFF scale is missed on Fig. 10."
(AC) Quality of the mentioned figures will be upgraded, and scale in Fig. 10 included for all subplots.

"11. Paragraph 2.1. Line 23: an homogenous -> a homogenous; Paragraph 2.3.1. Line 2: solution consists on inverting -> solution consists in inverting; Line 3: whose information is -> that are."

(AC) We will correct it and we will do our best to ensure that our English is easily readable and without mistakes.

**References**

[1] B Gaite et al. "Improving the location of induced earthquakes associated with an underground gas storage in the Gulf of Valencia (Spain)". In: *Physics of the Earth and Planetary Interiors* 254 (2016), pp. 46–59.

[2] S Cesca et al. "The 2013 September–October seismic sequence offshore Spain: a case of seismicity triggered by gas injection?" In: *Geophysical journal international* 198.2 (2014), pp. 941–953.

[3] SG Wesnousky et al. "Earthquake frequency distribution and the mechanics of faulting". In: *Journal of Geophysical Research: Solid Earth* 88.B11 (1983), pp. 9331–9340.

[4] DP Schwartz and KJ Coppersmith. "Fault behavior and characteristic earthquakes: Examples from the Wasatch and San Andreas fault zones". In: *Journal of Geophysical Research: Solid Earth* 89.B7 (1984), pp. 5681–5698.

[5] H Perea. "Falles actives i perillositat sísmica al marge nord-occidental del solc de Valencia". PhD thesis. Universitat de Barcelona, Barcelona: 382 p.(http://hdl. handle. net/10803/1919), 2006.

[6] JA Álvarez Gómez and F Martín González. "Una aproximación multidisciplinar al estudio de las fallas activas, los terremotos y el riesgo sísmico: 2ª Reunión Ibérica sobre Fallas Activas y Paleosismología: celebrada del 22 al 24 de octubre de 2014, en Lorca". In: (2014).

[7] Y Mukuhira et al. "Pore pressure behavior at the shut-in phase and causality of large induced seismicity at Basel, Switzerland". In: *Journal of Geophysical Research: Solid Earth* (2016).

[8] Y Mukuhira et al. "Characteristics of large-magnitude microseismic events recorded during and after stimulation of a geothermal reservoir at Basel, Switzerland". In: *Geothermics* 45 (2013), pp. 1–17.